# The dimeric Golgi protein Gorab binds to Sas6 as a monomer to mediate centriole duplication

Agnieszka Fatalska[1,2,3†]*, Emma Stepinac[4†], Magdalena Richter[1†‡], Levente Kovacs[1,2], Zbigniew Pietras[3], Martin Puchinger[5], Gang Dong[4], Michal Dadlez[3], David M Glover[1,2]*

[1]Department of Genetics, University of Cambridge, Cambridge, United Kingdom; [2]Division of Biology and Biological Engineering, California Institute of Technology, Pasadena, United States; [3]Institute of Biochemistry and Biophysics, Polish Academy of Sciences, Warsaw, Poland; [4]Department of Medical Biochemistry, Max Perutz Labs, Medical University of Vienna, Vienna, Austria; [5]Department of Structural and Computational Biology, Max Perutz Labs, University of Vienna, Vienna, Austria

**Abstract** The duplication and ninefold symmetry of the *Drosophila* centriole requires that the cartwheel molecule, Sas6, physically associates with Gorab, a trans-Golgi component. How Gorab achieves these disparate associations is unclear. Here, we use hydrogen–deuterium exchange mass spectrometry to define Gorab's interacting surfaces that mediate its subcellular localization. We identify a core stabilization sequence within Gorab's C-terminal coiled-coil domain that enables homodimerization, binding to Rab6, and thereby trans-Golgi localization. By contrast, part of the Gorab monomer's coiled-coil domain undergoes an antiparallel interaction with a segment of the parallel coiled-coil dimer of Sas6. This stable heterotrimeric complex can be visualized by electron microscopy. Mutation of a single leucine residue in Sas6's Gorab-binding domain generates a Sas6 variant with a sixteenfold reduced binding affinity for Gorab that cannot support centriole duplication. Thus, Gorab dimers at the Golgi exist in equilibrium with Sas6-associated monomers at the centriole to balance Gorab's dual role.

*For correspondence: af589@cam.ac.uk (AF); dmglover@caltech.edu (DMG)

†These authors contributed equally to this work

Present address: ‡Astra Zeneca, Cambridge, United Kingdom

Competing interests: The authors declare that no competing interests exist.

## Introduction

Centrioles are the ninefold symmetrical microtubule arrays found at the core of centrosomes, the bodies that organize cytoplasmic microtubules in interphase and mitosis. Centrioles also serve as the basal bodies of both non-motile and motile cilia, and flagellae. The core components of centrioles and the molecules that regulate their assembly are highly conserved (*Brito et al., 2012*). The initiation of centriole duplication first requires that the mother and daughter pair of centrioles at each spindle pole disengage at the end of mitosis. Plk4 then phosphorylates Ana2 (*Drosophila*)/STIL (*human*) at its N-terminal part, which promotes Ana2 recruitment to the site of procentriole formation, and at its C-terminal part, which enables Ana2 to bind and thereby recruit Sas6 (*Dzhindzhev et al., 2017*; *Dzhindzhev et al., 2014*; *McLamarrah et al., 2018*; *Ohta et al., 2014*). The ensuing assembly of a ninefold symmetrical arrangement of Sas6 dimers provides the structural basis for the ninefold symmetrical cartwheel structure at the procentriole's core. Sas6 interacts with Cep135 and in turn with Sas4 (*Drosophila*)/CPAP (*human*), which provides the linkage to centriole microtubules (*Hiraki et al., 2007*; *Jerka-Dziadosz et al., 2010*; *Kohlmaier et al., 2009*; *Lin et al., 2013*; *Roque et al., 2012*; *Schmidt et al., 2009*; *Tang et al., 2009*).

We recently identified an unexpected requirement for the protein, Gorab, to establish the ninefold symmetry of centrioles (*Kovacs et al., 2018*). Flies lacking Gorab are uncoordinated due to

basal body defects in sensory cilia, which lose their ninefold symmetry, and also exhibit maternal effect lethality due to failure of centriole duplication in the syncytial embryo. Gorab is a trans-Golgi-associated protein. Its human counterpart is mutated in the wrinkly skin disease, gerodermia osteodysplastica (*Burman et al., 2010*; *Burman et al., 2008*; *Di et al., 2003*; *Hennies et al., 2008*). By copying a missense mutation in gerodermia patients that disrupts the association of Gorab with the Golgi, we were able to create mutant *Drosophila* Gorab also unable to localize to trans-Golgi. However, this mutant form of Gorab was still able to rescue the centriole and cilia defects of *gorab* null flies. We also found that expression of C-terminally tagged Gorab disrupts Golgi functions in cytokinesis of male meiosis, a dominant phenotype that can be overcome by a second mutation preventing Golgi targeting (*Hiraki et al., 2007*; *Lin et al., 2013*). Thus, centriole and Golgi functions of *Drosophila* Gorab are separable.

The Golgi apparatus both delivers and receives vesicles to and from multiple cellular destinations and is also responsible for modifying proteins and lipids. Gorab resembles a group of homodimeric rod-like proteins, the golgins, which function in vesicle tethering (*Short et al., 2005*). The golgins associate through their C-termini with different Golgi domains, and their N-termini both capture vesicles and provide specificity to their tethering (*Gillingham and Munro, 2016*). There is known redundancy of golgin function, reflected by the overlapping specificity of the types of vesicles they capture (*Wong and Munro, 2014*). Gorab is rapidly displaced from the *trans*-side of the Golgi apparatus by Brefeldin A, suggesting that its peripheral membrane association requires ARF-GTPase activity (*Egerer et al., 2015*).

Previous studies of human Gorab indicated its ability to form a homodimer in complex with Rab6 and identified its putative coiled-coil region as a requirement to localize at the trans-Golgi (*Egerer et al., 2015*; *Witkos et al., 2019*). Our own studies on its *Drosophila* counterpart supported Gorab's ability to interact with itself, potentially through the predicted coiled-coil motif. However, we also found this region to overlap with the region required for Gorab's interaction with Sas6 (*Kovacs et al., 2018*). These findings raised the questions of how Gorab's putative coiled-coil region could facilitate interactions with the Golgi, on the one hand, and its Sas6 partner, on the other. To address this, we have employed hydrogen–deuterium exchange (HDX) in conjunction with mass spectrometry (MS). HDX enables the identification of dynamic features of protein by monitoring the exchange of main chain amide protons to deuteria in solution. Here, we have used HDX-MS to monitor the retarded exchange of amide protons localized between interacting regions of Gorab and Sas6 to identify the interacting surfaces within the Gorab–Sas6 complex. Together with other biophysical characterizations, this has revealed that Gorab is able to form a homo-dimer through its coiled-coil region but that it interacts as a monomer with the C-terminal coiled-coil of Sas6. Mutation of a critical amino acid in Sas6's Gorab-binding domain generates a variant of Sas6 with a sixteenfold reduced binding affinity for Gorab that is no longer able to support centriole duplication.

## Results

### Gorab dimerizes through its C-terminus to achieve Golgi localization

A previous study has identified a Golgi-targeting domain region of human Gorab from amino acids (aa) 200–277 (*Egerer et al., 2015*). This region, comprising predominantly a putative coiled-coil sequence, corresponds to aa 246–323 of *Drosophila* Gorab. To determine whether this region conferred the ability to homodimerize a characteristic of the golgins, we first wished to determine the oligomeric state of Gorab in solution. To this end, we expressed N-terminally MBP-tagged *Drosophila melanogaster* Gorab in *Escherichia coli*, affinity purified the recombinant protein on amylose resin, and carried out size exclusion chromatography coupled with multiangle light scattering (SEC-MALS) to determine its molecular mass (Mw) (*Figure 1A*). Whereas the theoretical mass of monomeric MBP-Gorab is 79.4 kDa, SEC-MALS indicated the Mw of the protein eluting in the major peak as 146.1 kDa. Thus, similar to other golgins, MBP-Gorab behaves as a dimer in solution.

To determine regions within Gorab that might be structured to support its homodimerization, we then performed HDX-MS on Gorab by incubation in $D_2O$ buffer for varying periods of time. The HDX pattern of Gorab after 10 s incubation with $D_2O$ buffer indicated almost complete exchange of hydrogen with deuterium in the N-terminal (aa 1–200) and very C-terminal parts (aa 318–338) of the protein, indicating high flexibility (*Figure 1B, Figure 1—figure supplement 1A*). Restricted

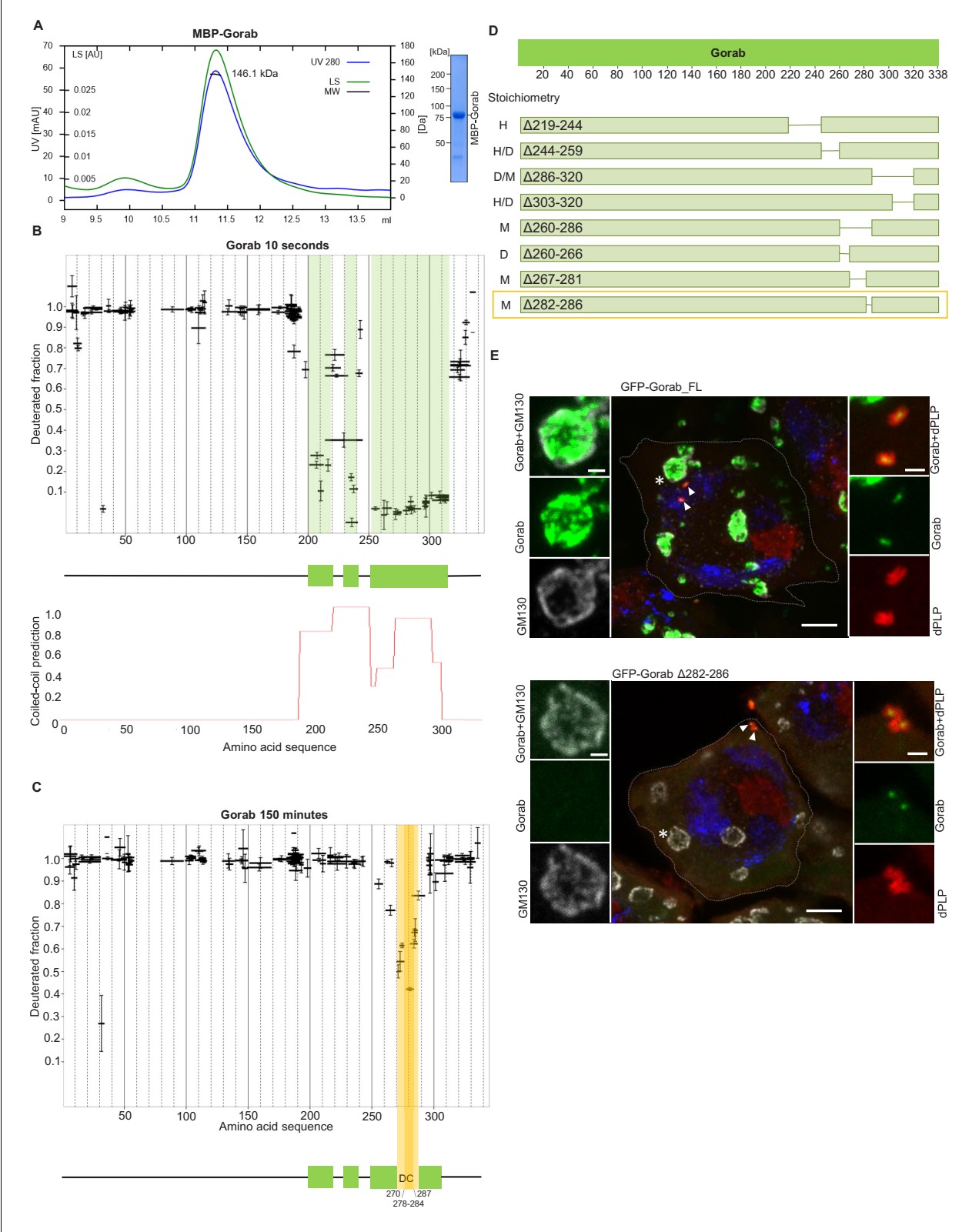

**Figure 1.** Gorab dimerizes through its C-terminal part to achieve Golgi localization. (A) Size exclusion chromatography-multiple angle light scattering (SEC-MALS) of Gorab. Blue: absorbance at 280 nm; green: light scattering (LS); short black line: molecular mass (Mm) of MBP-Gorab. Mm of MBP-Gorab monomer, 79.4 kDa. SDS-PAGE showing full-length MBP-Gorab after SEC. (B) Pattern of hydrogen–deuterium exchange (HDX) in Gorab peptides following 10 s incubation with deuterium oxide (heavy water). Black bars on Woods plots represent proteolytic peptides identified by mass

*Figure 1 continued on next page*

*Figure 1 continued*

spectrometry (MS) and positioned in relation to the Gorab amino acid sequence. The Y-axis shows fraction of deuteration compared to maximum level of measured deuteration. Mean of three experiments is shown. Error bars represent standard deviations. The N-terminal part of Gorab is highly flexible, whereas its C-terminal part has three protected regions (green blocks) that align with coiled-coil regions predicted by COILS (*Lupas et al., 1991*) (C) HDX pattern after 150 min incubation with $D_2O$. The region between aa 270–287 (highlighted in yellow) retains highly protected, and the region aa 278–284 with the highest protection is termed the dimerization core (DC). (D) Schematic showing outcome of SEC-MALS with Gorab having indicated deletions. H: higher-order structure; D: dimer; and M: monomer. Red box surrounds Gorab variant with shortest deletion (Δ282–286) that disrupts dimer formation corresponding to most protected region identified by HDX-MS (yellow). SEC-MALS data for each construct is presented in *Figure 1—figure supplement 2* (E) Localization of full-length and Δ282–286 Gorab (green) in G2 phase primary spermatocytes of adult males. N-terminally GFP-tagged Gorab transgenes were inserted in the same genomic location to ensure comparable expression by a constitutive poly-ubiquitin promoter. Immunostainings were performed with anti-GM130 (cis-Golgi marker, gray) and anti-dPLP (centrosome marker, red) antibodies. DAPI staining (blue) identifies the characteristic three-lobed nucleus of primary spermatocytes prior to meiosis. The dashed line outlines the border of a single spermatocyte. Golgi, indicated by asterisks, and centrosomes, by arrowheads, are shown in insets. In total, 30 primary spermatocytes from four transgenic testes expressing full-length Gorab and 32 primary spermatocytes from four testes expressing Gorab Δ282–286 were imaged. All showed the same Gorab distribution. Scale bar: 5 µm; scale bar in insets: 1 µm.

The online version of this article includes the following source data and figure supplement(s) for figure 1:

**Figure supplement 1.** Dimerization of Gorab through interactions between the coiled-coil domains in its C-terminal region.

**Figure supplement 2.** Requirements of regions of the Gorab coiled-coil region for dimerization as revealed by size exclusion chromatography (SEC) coupled with multiangle light scattering of Gorab deletions.

**Figure supplement 3.** Gorab Δ282–286 localizes to centrioles but not to the Golgi in mitotic cells of imaginal discs.

**Figure supplement 3—source data 1.** Percentage of flies climbing 5 cm in 15 s in three independent biological replicates of the indicated genotypes.

deuterium exchange in the C-terminal part suggested three distinct protected, structured regions (aa 200–220, 230–240, and 252–315) matching the coiled-coil predictions by the COILS program (*Lupas et al., 1991*). This suggests that the dimerization of Gorab might be driven by coiled-coil interactions on its C-terminus (*Figure 1B*). SEC-MALS of two Gorab C-terminal truncations (aa 191–338 and 191–279) confirmed the importance of the aa 279–338 region in the dimer formation (*Figure 1—figure supplement 1B*). In the HDX experiment, longer periods of incubation identified regions permitting 'breathing' of this secondary structure (*Figure 1C, Figure 1—figure supplement 1A*), such that after 150 min only the region between aa 270–287, and especially aa 278–284, remained protected from the deuterium exchange. The high protection shown by this region is most likely to reflect regions of stability around the core of Gorab's dimerization domain (*Figure 1C*).

It is also possible that the regions of stability detected in the above experiment may be caused by allosteric changes coming from interactions elsewhere. Therefore, to test the relevance of the aa 270–287 fragment for homodimerization, we designed a set of MBP-Gorab constructs harboring deletions within the C-terminal part and determined their molecular mass using SEC-MALS. This revealed that the aa 260–286 region is essential for dimer formation (*Figure 1D, Figure 1—figure supplement 2*) in accord with the findings of HDX-MS. Moreover, SEC-MALS data revealed that deletions within other parts of the C-terminal domain disrupt the dimeric structure of MBP-Gorab and result in the formation of higher-order structures, or mixtures of these with monomers (*Figure 1D, Figure 1—figure supplement 2*). This suggests that the entire C-terminal domain (aa 200–320) is organized in such a way to permit dimerization and that even minor changes within the sequence lead to structural instability.

To test the consequence of disrupting dimerization in vivo, we created a transgene encoding Gorab with a 282–286 aa deletion. We then generated transgenic flies with full-length (FL) Gorab or Gorab Δ282–286 under the control of constitutive poly-ubiquitin promoter, allowing moderately elevated expression of the given transgene. We used a site-specific integrase system to integrate them into the same genomic location so that they would be expressed at a similar level. An eGFP tag on the N-terminus of the transgenic protein allowed us to determine its subcellular localization in primary spermatocytes and larval imaginal discs. In agreement with our earlier findings (*Kovacs et al., 2018*), the vast majority of wild-type eGFP-Gorab localized in the position of the trans-Golgi, inside the Golgi cisternae and surrounded by the GM130 cis-Golgi marker, in large primary spermatocytes (*Figure 1E*). Accordingly, the majority of FL eGFP-Gorab colocalized with Golgin-245 trans-Golgi marker as we observed in primary spermatocytes and wing imaginal discs and, in accord with our

previous findings, a small fraction of FL Gorab was recruited to centrosomes (*Figure 1E, Figure 1—figure supplement 3*). By contrast, Gorab Δ282–286 did not localize to the Golgi but retained its centrosomal localization both in primary spermatocytes and in wing disc cells (*Figure 1E, Figure 1—figure supplement 3*). In line with its centriolar localization, Ubq-Gorab Δ282–286 was able to fully rescue the coordination, viability, and fertility defects of *gorab* mutants (*Kovacs et al., 2018*). This indicates that dimerization of Gorab is required for its trans-Golgi localization but not for its centriolar localization and function.

## Gorab interacts with Sas6 through its C-terminus and forms a heterotrimeric complex

The above findings indicated that Gorab localizes to the centrosome as monomer, leading us to investigate Gorab's oligomeric state in relation to its previously reported interaction with Sas6 (*Kovacs et al., 2018*). We therefore set out to reconstitute a complex between these two proteins in vitro that we could analyse using SEC and SEC-MALS (*Figure 2A, B*). SEC confirmed that when mixed in a molar ratio 1 part Gorab : 2 parts Sas6, a stable complex was formed (*Figure 2A*). MALS measurements indicated the molecular mass of this complex eluting in the major peak to be 258.7 kDa (±0.061%), which would correspond to a heterotrimer of homodimeric MBP-Sas6 and monomeric MBP-Gorab, whose theoretical mass is 272.2 kDa (*Figure 2B*). We also confirmed heterotrimeric complex formation with Sas6 and Gorab truncations (Sas6 CC [338–472 aa] + Gorab CC short [191–279 aa], and MBP-Sas6 CC short [404–463 aa] + MBP-Gorab CC short [191–279 aa]) using SEC-MALS (*Figure 2—figure supplement 1A*). This indicates that Gorab is a dimer in solution but binds Sas6 dimer as a monomer.

To determine the region of Gorab that interacts with Sas6, we performed two sets of HDX-MS measurements: Gorab alone and Gorab in complex with Sas6 (*Figure 2C, Figure 2—figure supplement 1B*). By analyzing the HDX patterns for Gorab, we could observe two distinct changes in deuteration levels. The 230–260 aa segment became slightly more stable when Gorab was in complex with Sas6 than when on its own (*Figure 2C*, pink box, *Figure 2—figure supplement 1B*). In contrast, the 270–315 aa segment appeared to be destabilized (*Figure 2c*, blue box, *Figure 2—figure supplement 1B*). It is most likely that the 230–260 aa segment of Gorab, which is stabilized upon Sas6 binding, represents the interaction surface between the two proteins. As Gorab appears to dissociate to a monomer during complex formation with Sas6, this suggests that the aa 270–315 segment at the dimerization core (*Figure 1C, D*) becomes destabilized. To test the importance of the region of Gorab indicated by HDX-MS to form an interaction surface with Sas6, we performed binding assays between Sas6 and a series of deletions and truncations of Gorab (*Figure 2D*). This revealed Gorab Δ244–259 aa as the shortest deletion that gives the maximal observed decrease in Sas6 binding. The flanking deletions, such as Δ219–244 aa and Δ260–286 aa, could still bind Sas6 but weakly. C-terminal truncations showed a weak decrease in binding only when they removed sequence between 260 and 270 aa and a strong decrease in binding if they removed residues between 246 and 260 aa. These findings accord with the HDX data and strongly suggest that the 244–260 aa region of Gorab binds strongly to Sas6 and that weaker interacting surfaces extend between aa 219–244 and aa 260–270 (*Figure 2D, Figure 2—figure supplement 2A, B*).

## Sas6 interacts with Gorab via its C-terminus

We then wished to determine the region within Sas6 that interacts with Gorab. To do so, we monitored HDX of Sas6 alone and when in complex with Gorab. We observed clear differences in deuteration level within 20 residues (aa 440–460) towards the C-terminus of Sas6 that showed increased stability when in complex with Gorab (*Figure 3A, Figure 3—figure supplement 1A*). We also observed less pronounced protection in peptides from other regions. However, our previous study showed that the Sas6 region aa 351–472 (*Kovacs et al., 2018*) is sufficient to bind Gorab. Moreover, we found that the Sas6 segment aa 404–463 forms a stable complex with the 191–279 aa segment of Gorab (*Figure 2—figure supplement 1A*). We therefore concluded that the Sas6 region aa 440–460, showing the highest protection from HDX upon Gorab binding, lies on the interaction surface. This led us to attempt to identify single residues within the aa 440–460 region that are essential for this interaction. A multiple sequence alignment of this region of Sas6 from various species (*Figure 3—figure supplement 1B*) identified highly conserved aa that we mutated to alanine residues

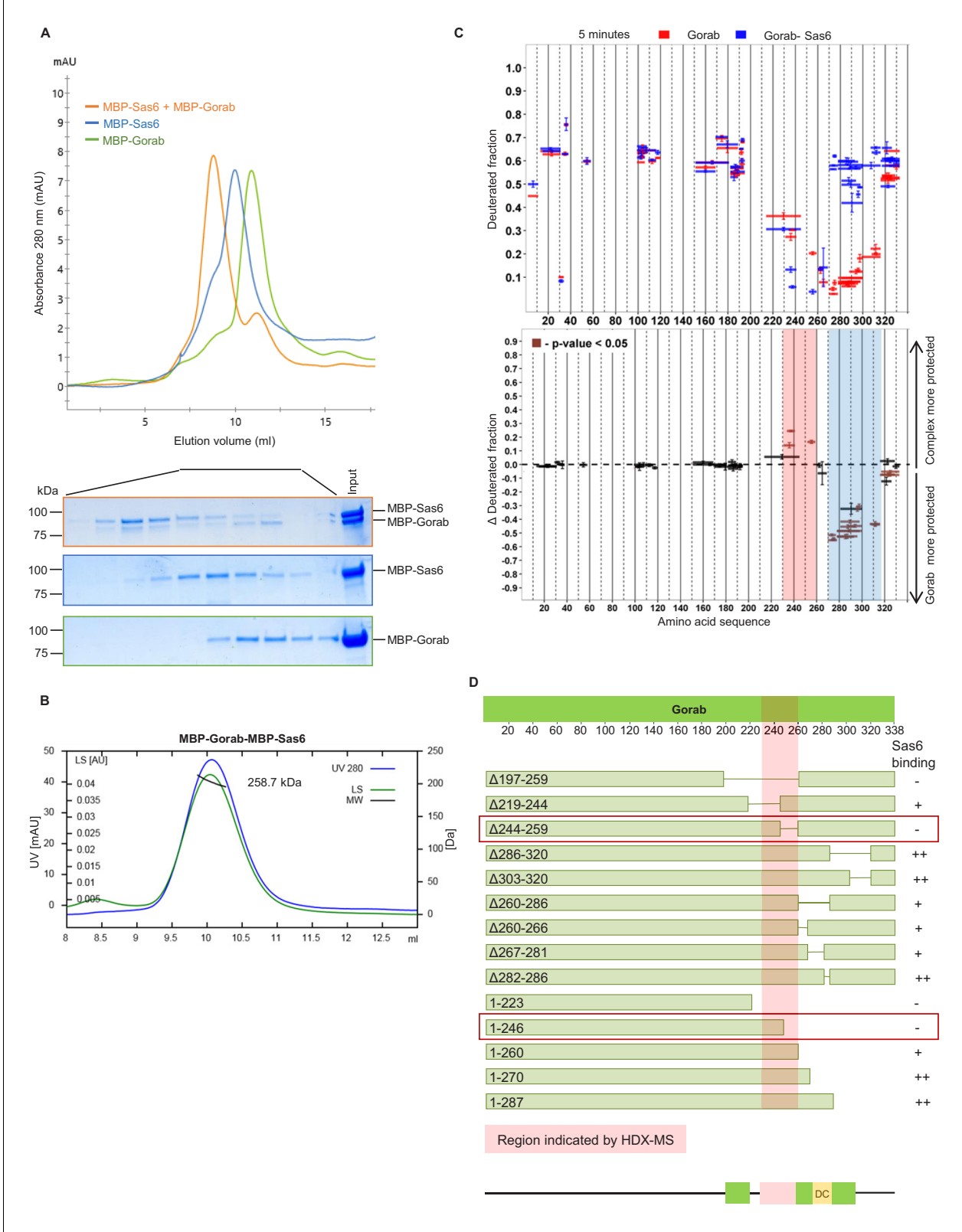

**Figure 2.** Gorab interacts with Sas6 through its C-terminal part and forms a heterotrimeric complex. (**A**) Size exclusion chromatography (SEC) of Gorab, Sas6, and Gorab–Sas6 complex. Green: absorbance 280 nm of MBP-Gorab; blue: absorbance 280 nm of MBP-Sas6; yellow: absorbance at 280 nm of MBP-Gorab-MBP-Sas6 complex. Lower panel: SDS-PAGE of SEC fractions. (**B**) Size exclusion chromatography coupled with multiangle light scattering of Gorab–Sas6 complex. Blue: absorbance at 280 nm; green: light scattering; black: molecular mass (Mm) of MBP-Gorab-MBP-Sas6 complex. Mm of

*Figure 2 continued on next page*

*Figure 2 continued*

MBP-Sas6 monomer is 96.4 kDa; Mm of MBP-Gorab monomer, 79.4 kDa. (C) Upper panel: hydrogen–deuterium exchange (HDX) pattern of Gorab in complex with Sas6 following 5 min incubation with D₂O. Gorab peptides alone (red bars) and when in complex with Sas6 (blue bars). X-axis: position of peptides in amino acid sequence; Y-axis: fraction of deuteration compared to maximum level of calculated deuteration. Mean of two experiments is shown. Error bars show both values measured. Lower panel: differences between deuteration of Gorab peptides alone and in complex with Sas6, derived by subtraction of deuteration levels shown in the upper panel. Brown bars indicate peptides for which the differences measured in repeated experiments satisfied the Welsh *t*-test with p<0.05. Blue region: peptides that are more protected from deuterium exchange when Gorab is not in the complex (aa 270–315); red region: peptides that are more protected when Gorab is in complex with Sas6 (aa 230–260). (D) Schematic of ability Gorab deletions/truncations to bind Sas6 in vitro; –: no binding; +: binding; ++: strong binding. Red highlighted box: region identified by HDX-MS (aa 230–260). Red boxes: region of Gorab essential for Sas6 binding (aa 244–260). Individual results for each construct are presented in *Figure 2—figure supplement 2A, B*.

The online version of this article includes the following source data and figure supplement(s) for figure 2:

**Source data 1.** Uncropped SDS-PAGE corresponding to *Figure 2A*.
**Figure supplement 1.** Regions of Gorab required for Sas6 binding.
**Figure supplement 2.** Consequences of deletion or truncation of Gorab for Sas6 binding.

---

(M440A, L447A, S452A, L456A). We then assessed the consequences for interaction with Gorab by in vitro binding assays (*Figure 3B, Figure 3—figure supplement 2A*) and SEC (*Figure 3—figure supplement 2B*). These experiments indicated that Sas6 L447A was not able to bind Gorab in vitro.

To further confirm this, we next performed fluorescence correlation spectroscopy (FCS), enabling us to measure the change of diffusion times and therefore diffusion coefficients of one fluorescently labeled molecule upon binding to its non-fluorescent binding partner. In this case, we mixed the fluorescently labeled MBP-Gorab coiled-coil fragment (aa 191–279) with non-fluorescent MBP-wild type Sas6 or the Sas6 L447A mutant, measured the change in the diffusion time with increasing Sas6 concentrations, and determined the dissociation constant ($K_D$) of the Gorab–Sas6 binding (*Figure 3C*). To control for the ability of wild-type Sas6 dimers to oligomerize through their head-to-head interactions, we also tested Gorab binding to the Sas6-F143D mutant, which abolishes higher oligomerization and found no significant differences in the outcome (*Figure 3—figure supplement 3A*). Our measurements revealed that the $K_D$ of Gorab's interaction with wild-type Sas6 was in the low nanomolar range of 47 nM, but that the binding strength to Sas6 L447A mutant was 16 times weaker ($K_D$ = 798 nM). We next investigated the impact of mutations in the aa 440–460 region of Sas6 on its ability to self-associate. We found that Sas6-wild type, Sas6-M440A, Sas6-L447A, Sas6-S452A, and Sas6-L456A all displayed the same elution profile in SEC (*Figure 3—figure supplement 3B*). This indicates that all of the Sas6 point mutants behave like Sas6-wild type in being able to form homodimers. Moreover, L447, which is located at the C-terminal region of the coiled coil of Sas6, is predicted to lie at position 'a' of the 'a-g' heptad repeats with M440 lying in a similar position (*Figure 3—figure supplement 4A, B*). Yet L447A prevents Gorab binding, whereas M440A does not. Our modeling analysis using CCBuilder 2.0 (*Wood and Woolfson, 2018*) predicts that mutants M440A and L447A have a BUDE energy of –717 and –714, respectively, which is very similar to that of Sas6-wild type (−724) (*Figure 3—figure supplement 4C*). This is consistent with the confirmed dimeric state of all the Sas6 constructs (*Figure 3—figure supplement 3B*). Notably, however, residue L447 is closer to the C-terminus of the predicted coiled-coil of Sas6 than M440, and the downstream two heptads following L447 may not hold the dimer tightly together as only one of the residues at positions 'a' and 'd' of these two heptads, that is, I457, is hydrophobic. We therefore conclude that the L447A does not prevent Sas6 from forming a homodimer but the local structure around L447 might be partially perturbed in the mutant L447A. Such subtle change in the local region seems to be sufficient to prevent the interaction of Sas6-L447A with Gorab.

To determine the consequences of the Sas6-L447A mutation for interactions with Gorab in vivo, we created transgenic flies expressing either wild-type Sas6 or the Sas6-L447A mutant under the control a GAL4-inducible UAS regulatory sequence. We also generated transgenes of Sas6-M440A as this mutation is predicted to lie adjacent to L447 in the Sas6 coiled coil but does not affect Gorab-binding ability. The transgenes were integrated into the same genomic location using a site-specific integrase system to ensure their expression at comparable levels. The expression of these transgenes was induced either by neuronal elav-GAL4 or by ubiquitous Act5C-GAL4 driver in a *sas6 null* mutant background. We first assessed the ability of Sas6-L447A to rescue the coordination

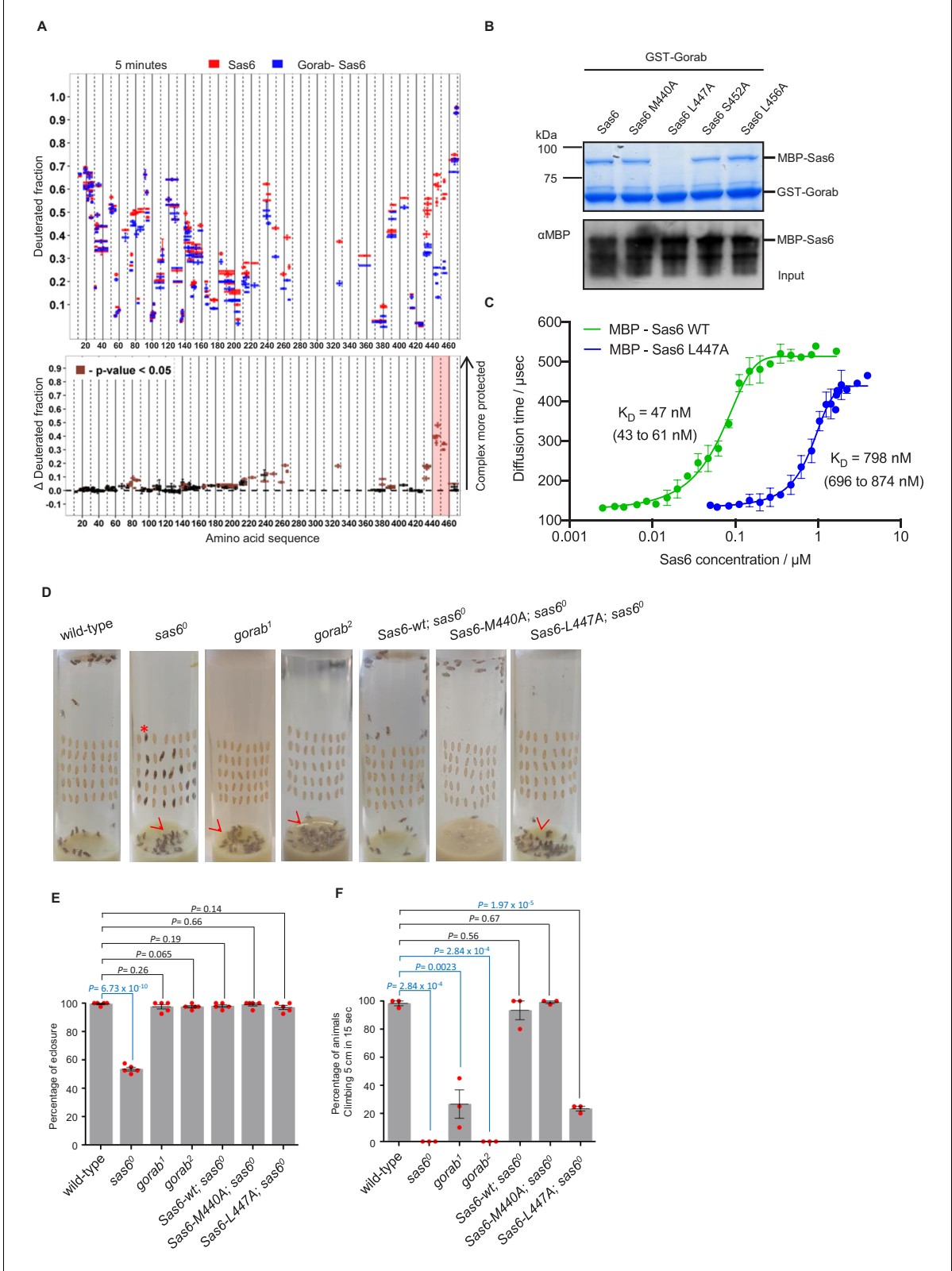

**Figure 3.** Sas6 interacts with Gorab through its C-terminal region. (**A**) Upper panel: hydrogen–deuterium exchange pattern of Sas6 in complex with Gorab following 5 min incubation with $D_2O$. Sas6 peptides alone (red bars) and when in complex with Gorab (blue bars). X-axis: position of peptides in amino acid sequence; Y-axis: fraction of deuteration compared to maximum level of calculated deuteration. Mean of two experiments is shown. Error bars show both values measured. Lower panel: differences between deuteration of Sas6 peptides alone and in complex with Gorab, derived by

*Figure 3 continued on next page*

*Figure 3 continued*

subtraction of deuteration levels shown in the upper panel. Brown bars indicate peptides for which the differences measured in repeated experiments satisfied the Welsh *t*-test with p<0.05. Red box: peptides protected most from exchange when Sas6 is in complex with Gorab (aa 440–460). (B) Pull-down assay for wild type (WT) and point mutants of Sas6 with Gorab. Upper panel: SDS-PAGE of the binding assay in which Gorab is the bait and WT and point mutants of Sas6 are the prey. Lower panel: western blot showing input of WT and point mutants of Sas6. Leucine 447 in Sas6 is essential for Gorab binding. (C) Fluorescence correlation spectroscopy measurements of fluorescently labeled MBP-Gorab (aa 191–279) binding non-labeled MBP-Sas6 WT (green) or MBP-Sas6 L447A mutant (blue). Mean values of the dissociation constants for Gorab–Sas6 WT and Gorab–Sas6 L447A are 47 and 798 nM, respectively, with 95% confidence intervals (brackets). Error bars show standard deviation of three independent measurements. (D) Eclosion phenotype of indicated mutants and *sas6º* null flies expressing the indicated transgenes. Pupae (40/vial) were aligned on the side of the vial and left to eclose. Asterisk exemplifies an individual that died at the pharate adult stage within the pupal case. Arrowheads point to eclosed but uncoordinated adults stuck in media. Expression of *Sas6-wild-type*, *Sas6-M440A*, and *Sas6-L447A* constructs was induced in sas6º neurons by the *elav-GAL4* driver. Flies were raised and the experiments were performed at 25°C. Ubiquitous expression of the constructs driven by *Act5C-GAL4* gave similar eclosion rates (not shown) (E). Quantification of eclosion rate of flies. Datapoints represent percentage of eclosed adults from each replica. Means and standard errors are shown for N = 5 independent biological replicates per genotype; n = 40 flies investigated in each replica. p-Values of two-tailed, unpaired *t*-tests are shown. p-Value in blue indicates significant difference (95% confidence interval). (F) Climbing assays of indicated mutants and *sasº* flies expressing the indicated transgenes. The expression of *Sas6-WT*, *Sas6-M440A*, and *L447A* rescue constructs was induced in sas6º null neurons by the *elav-GAL4* driver. Ubiquitous expression from the *Act5C-GAL4* driver gave similar eclosion rates (not shown). Cohorts of 15 flies raised at 25°C were scored for the number of individuals able to climb 5 cm in 15 s after being tapped down to bottom of vial. Means and standard errors are shown for N = 3 independent experiments per genotype; n = 15 flies investigated in each experiment. p-Values of two-tailed, unpaired t-tests are shown. p-Value in blue indicates significant difference (95% confidence interval).

The online version of this article includes the following source data and figure supplement(s) for figure 3:

**Source data 1.** Table representing three independent fluorescence correlation spectroscopy (FCS) measurements for each of the complex formations: between Gorab and Sas6 wild type (WT); and Gorab and Sas6 L447A.

**Source data 2.** Percentage of flies eclosed in five independent biological replicates of the indicated genotypes.

**Figure supplement 1.** Regions of Sas6 interacting with Gorab.

**Figure supplement 2.** Consequences of point mutations in Sas6 for its interactions with Gorab.

**Figure supplement 2—source data 1.** Uncropped SDS-PAGE corresponding to *Figure 3—figure supplement 2B*.

**Figure supplement 3.** Sas6–Gorab interactions.

**Figure supplement 3—source data 1.** Table representing three independent fluorescence correlation spectroscopy (FCS) measurements of complex formation between Gorab and Sas6 F143D.

**Figure supplement 4.** Structural analyses of the coiled coil of Sas6.

---

phenotypes of the *sas6 null* mutant raised at 25°C (*Figure 3D–F*). Pharate *sas6 null* adults are unco-ordinated to the extent that only about 50% of pupae can eclose from the pupal case and the ones that do emerge are unable to complete our climbing test (to scale within 15 s). These mutant pheno-types were completely rescued by the elevated expression of wild-type or M440A Sas6 transgenes (*Figure 3D–F*). In contrast, Sas6 L447A was able to rescue the eclosion phenotype of the *sas6 null* and could only partially the coordination phenotype such that 30% of these flies were able to pass the climbing test (*Figure 3D–F, Video 1*). If this reflected failure of Sas6-L447A to bind Gorab, then the phenotype should be similar to that of *gorab¹* and *gorab²* mutants. When *gorab¹* and *gorab²* flies are raised at 29°C, both mutant alleles display similarly severe uncoordination defects and the flies are completely unable to climb. At 25°C, however, *gorab²* flies showed a similarly severe total loss of coordination, whereas *gorab¹* flies were moderately uncoordinated with around 25% being able to climb 5 cm in 15 s (*Figure 3F*). Flies with heteroallelic combinations of *gorab¹* and *gorab²* showed intermediate phe-notypes, more similar to *gorab²*. Thus the extent of rescue of the coordination phenotype of the *sas6 null* by overexpression of Sas6 L447A was reminiscent of the *gorab¹* hypomorph (*Figure 3F, Videos 1* and *2*) and in accord with

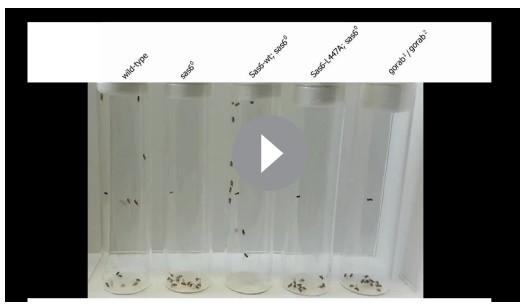

**Video 1.** Climbing test of *sas6º* flies with and without indicated Sas6 variant transgenes. Flies were transferred into assay vials without anesthesia and let them accommodate for 15 min, after which they were tapped down, and their climbing ability was assessed. https://elifesciences.org/articles/57241#video1

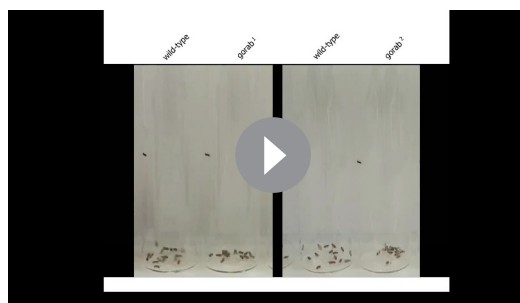

**Video 2.** Climbing test of *gorab* mutant flies. Flies were transferred into assay vials without anesthesia and let them accommodate for 15 min, after which they were tapped down, and their climbing ability was assessed.

https://elifesciences.org/articles/57241#video2

the greatly reduced ability of Sas6-L447A to bind Gorab.

To analyze the relative consequences of the above mutations on centrosome number, we used late third larval instar wing discs as a model. The wing pouch part of the disc (the region that will form the wing blades of adults) is formed by columnar epithelial cells, in which interphase centrioles are apically localized and can be imaged and quantified on a single plane (*Figure 4A*). As the cells of this layer enter mitosis, they round up and move deeper within the epithelium allowing mitotic centrioles to be imaged at the spindle poles on a different plane (*Figure 4A*). We found that the elevated expression of both wild-type Sas6 and the Sas6-M440A transgene from the Act5C-GAL4 driver fully rescued the centriole loss of *sas6* null larvae in interphase cells (*Figure 4—figure supplement 1*). By contrast, Sas6 L447A gave only partial rescue restoring only 8% of centriole numbers. This compares to the *gorab²* null mutant in which centrosomes are absent and the *gorab¹* hypomorph, in which centrosome numbers are reduced to around 35% of wild type. Rescue of the *sas6* null by Sas6-L447A restored centrosome numbers to a similar level as in the heteroallelic *gorab¹/gorab²* combination, which is strongly hypomorphic (*Figure 4—figure supplement 1*). We found a similar distribution of centrosomes in the mitotic wing disc cells of the above flies (*Figure 4B*). Whereas expression of wild-type Sas6 or Sas6-M440A totally rescued the complete mitotic centrosome loss of the *sas6* null, expression of Sas6-L477A resulted in partial rescue in which we observed several phenotypic categories (*Figure 4B*): (1) mitotic cells with two normal centrosomes, (2) cells with a single normal centriole, (3) cells with two centrioles with one showing reduced asterless (Asl) and absent Rcd4 staining (*Figure 4—figure supplement 1C*), (4) cells with two centrioles with both showing such reduced staining, (5) cells with only one centriole having reduced staining, and (6) cells with no centrioles. The distribution of cells exhibiting these phenotypes indicated that expression of Sas6 L447A in the *sas6* null background resulted in a phenotype most similar to that of heteroallelic *gorab¹/gorab²* flies or *gorab²* null flies. Thus, the consequences of Sas6-L447A upon centrosome number accord with the protein having reduced ability to bind Gorab.

If the above interpretation is correct, then Sas6-M440A should still able to recruit Gorab to the centrosome in vivo, whereas Sas6-L447A should not. This led us to determine how Sas6-L447A would affect Gorab localization in wing discs. To this end, we generated fly lines giving constitutive expression of GFP-tagged wild-type Gorab from the poly-ubiquitin promoter and wild-type Sas6, Sas6-M440A, or Sas6-L447A from the UAS promoter driven by Act5C-GAL4. We then stained wing discs to reveal the centriole protein Asl and monitored localization of GFP-Gorab in both interphase and mitotic centrosomes (*Figure 4C*). Asl staining revealed centrosomes to be absent in *sas6* null discs and rescued by either wild-type Sas6 or Sas6-M440A in both interphase and mitotic cells (*Figure 4C*). In all cases, Gorab could be detected on these rescued centrosomes. We were, however, unable to detect Gorab on the rare centrosomes present following expression of Sas6-L447A in the *sas6* null background irrespective of the intensity of Asl or Rcd4 staining (*Figure 4—figure supplement 1C*, n = 243 centrioles counted in total). Thus, Sas6-L447A binds Gorab extremely poorly in vitro and is unable to recruit Gorab onto the centriole in vivo, a property that is associated with dramatically reduced centriole duplication even though Sas6 L447A is still able to localize to the centrioles (*Figure 4—figure supplement 1D*).

Together, these results show that the L447-dependent interaction of Sas6 with Gorab mediates Gorab's association with the centrosome, and as a consequence, the phenotype of *Sas6-L447A* strongly resembles that of a *gorab* strong hypomorph.

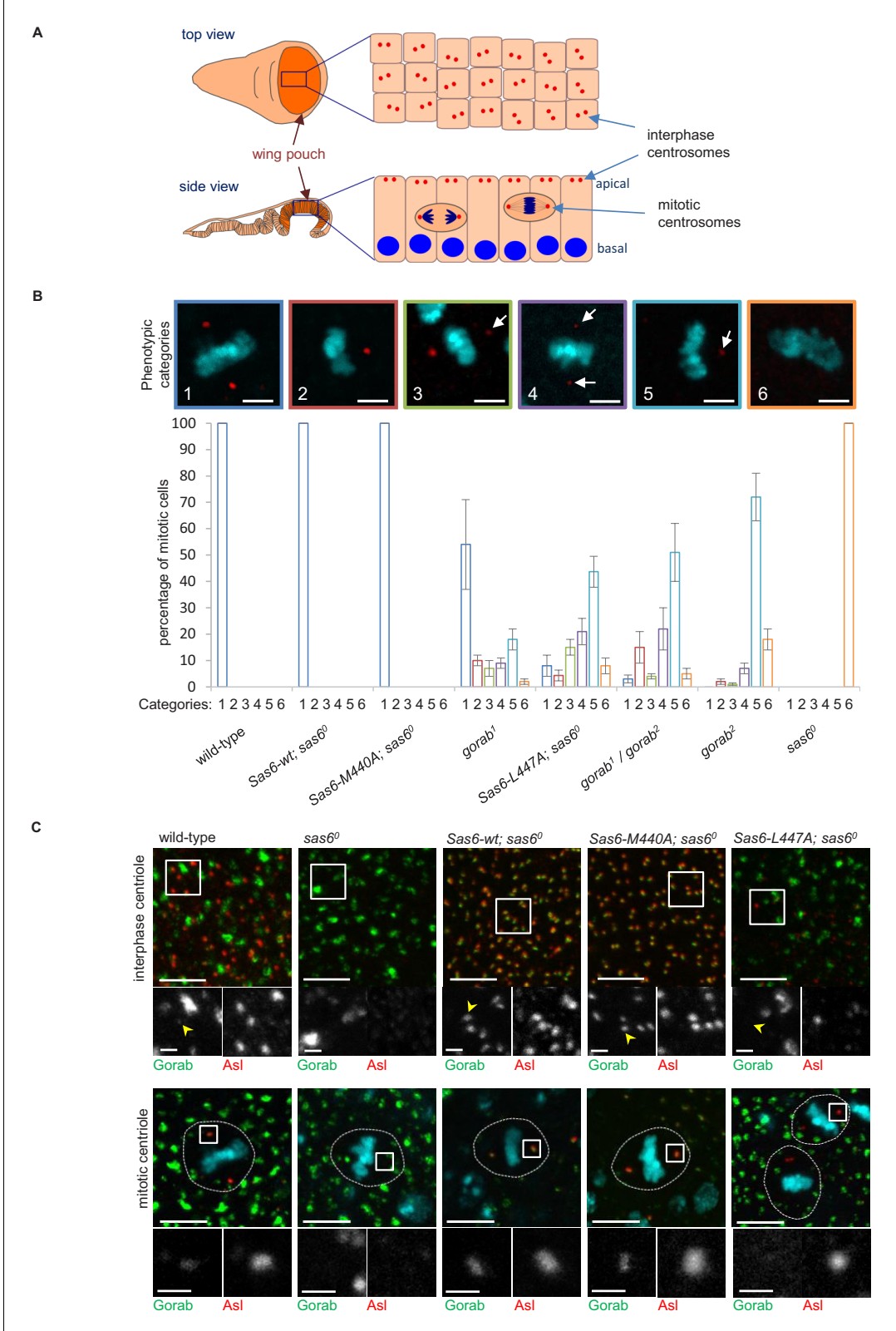

**Figure 4.** Consequences of Sas6 L447A upon Gorab recruitment. (**A**) Schematic showing the organization of centrosomes in columnar wing disc epithelia in wandering late third instar larvae. Centrosomes (red dots) localize to the apical surface of interphase cells, whereas nuclei (blue) have a basal localization. Mitotic cells round up and become localized between the apical and basal surfaces. Localization and quantification was performed on the middle part of the wing pouch (orange zone) of the disc. (**B**) Centriolar phenotype of the indicated *sas6* and *gorab* mutants and *sas6* mutants

*Figure 4 continued on next page*

*Figure 4 continued*

rescued with the indicated transgenes. The upper panel shows examples of the six phenotypic categories: 1: two normal centrioles; 2: one normal centriole; 3: one normal and one centriole showing diminished staining; 4: two centrioles with diminished staining; 5: one centriole with diminished staining; 6: no centrioles. Chromosomes are stained with DAPI (blue) and centrioles by anti-asterless (Asl) immunostaining (red). Arrows indicate centrioles with diminished Asl signal. Scale bar: 2.5 µm. The lower histograms show the quantification of phenotypic categories for the indicated genotypes. Colors correspond to phenotypic categories (also indicated numerically). Means and standard errors are indicated for three independent experiments, each assessing 100 mitotic cells from 12 wing discs from each genotype. (C) Localization of GFP-Gorab expressed from the constitutive poly-ubiqitin promoter in *sas6* null mutant wing discs in the absence and presence of the indicated Sas6 variants. The Sas6-WT (wild type), Sas6-M440A, and Sas6-L447A transgenes were integrated into the same genomic locus to achieve comparable expression levels from the ubiquitous Act5C-GAL4 driver. Interphase centrioles from the apical zone (upper row) and mitotic centrioles of metaphase cells (lower row) were visualized by anti-Asl immunostaining (red). GFP-Gorab not associated with centrioles corresponds to the Golgi fraction (see also *Figure 4—figure supplement 1D*). A total of 100 interphase cells and 30 mitotic cells from five independent wing discs were imaged for each genotype, all of which showed similar Gorab distributions within the same genotype. Scale bar: 5 µm; scale bar in insets: 1 µm.

The online version of this article includes the following source data and figure supplement(s) for figure 4:

**Source data 1.** Number of centrioles in different categories in wing imaginal disc of indicated genotypes.

**Figure supplement 1.** Centriolar phenotype in *gorab, sas6, and sas6* expressing various *Sas6* transgenes.

**Figure supplement 1—source data 1.** Number of centrioles counted in a 50 × 50 um wing disc area in indicated genotypes.

## Gorab undergoes an antiparallel interaction with Sas6

Having mapped the minimal regions essential for the interaction on both Gorab and Sas6, we wanted to determine the orientation of each of the protein molecules within the complex. To this end, we visualized the complex by rotary shadowing electron microscopy (EM), a method highly suited to reveal the elongated coiled coils present in Sas6. We generated three N-terminally MBP-tagged constructs of Gorab corresponding to the FL molecule, the putative coiled-coil domain alone (aa 191–338), and a C-terminally truncated form of the putative coiled-coil domain (aa 191–279) (*Figure 5A*). The resulting EM images, together with their respective schematic interpretations, are shown for FL Sas6 alone and Sas6 FL in complex with different MBP-Gorab constructs (*Figure 5B–E*, *Figure 5—figure supplement 1*).

The appearance of Sas6 FL molecules confirms previous findings (*Cottee et al., 2015*; *Kitagawa et al., 2011*; *Qiao et al., 2012*; *van Breugel et al., 2011*) that Sas6 exists as a parallel dimer in which the two globular head domains lie on one end of a coiled coil, visible as a long rod of mean length 41 nm (n = 106) (*Figure 5B*). When Sas6 FL was complexed to the different constructs of N-terminally MBP-tagged Gorab, we observed the two globular head domains of Sas6 and one globular domain of MBP to be separated by a long coiled-coil rod (*Figure 5C–E*). The mean length of the rod between the Sas6 head domains and the MBP tag was greater than the Sas6's coiled-coil rod alone; 54.8 nm (n = 87), 51.5 nm (n = 118), and 49.6 nm (n = 114) for Sas6 in complex with MBP-Gorab FL, MBP-Gorab CC long, and MBP-Gorab CC short, respectively (*Figure 5F*). These observations confirm that Sas6 forms a parallel dimer and indicate that a Gorab monomer binds Sas6 dimer in an antiparallel manner.

## Gorab interacts with Rab6 via its C-terminal coiled-coil domain

As the golgins are known to interact with Rab family proteins through their C-terminal regions (*Short et al., 2005*), we sought to determine whether Gorab's interaction with Rab6 followed similar requirements. To demonstrate the interaction between Gorab and Rab6, we performed an in vitro binding assay between Gorab and active (GTP locked mutant) Rab6 (*Figure 6A, Figure 6—figure supplement 1A*). In order to determine which part of Gorab was involved in the interaction with Rab6, we made a series of N- and C-terminal truncations of Gorab. This revealed that C-terminal region of Gorab (aa 223–338) is able to weakly bind Rab6. However, for a complete interaction with Rab6, a longer C-terminal region of Gorab (195–338 aa) is needed (*Figure 6B, Figure 6—figure supplement 1B*). This region encompasses Gorab's entire coiled-coil domain that participates in the homodimerization. As even minor changes in this region can disrupt the dimerization, and hence Rab6 binding, we were not able to map the binding surface through this type of approach. Thus, the ability for Gorab to form a dimer appears critical for complex formation between Gorab and Rab6.

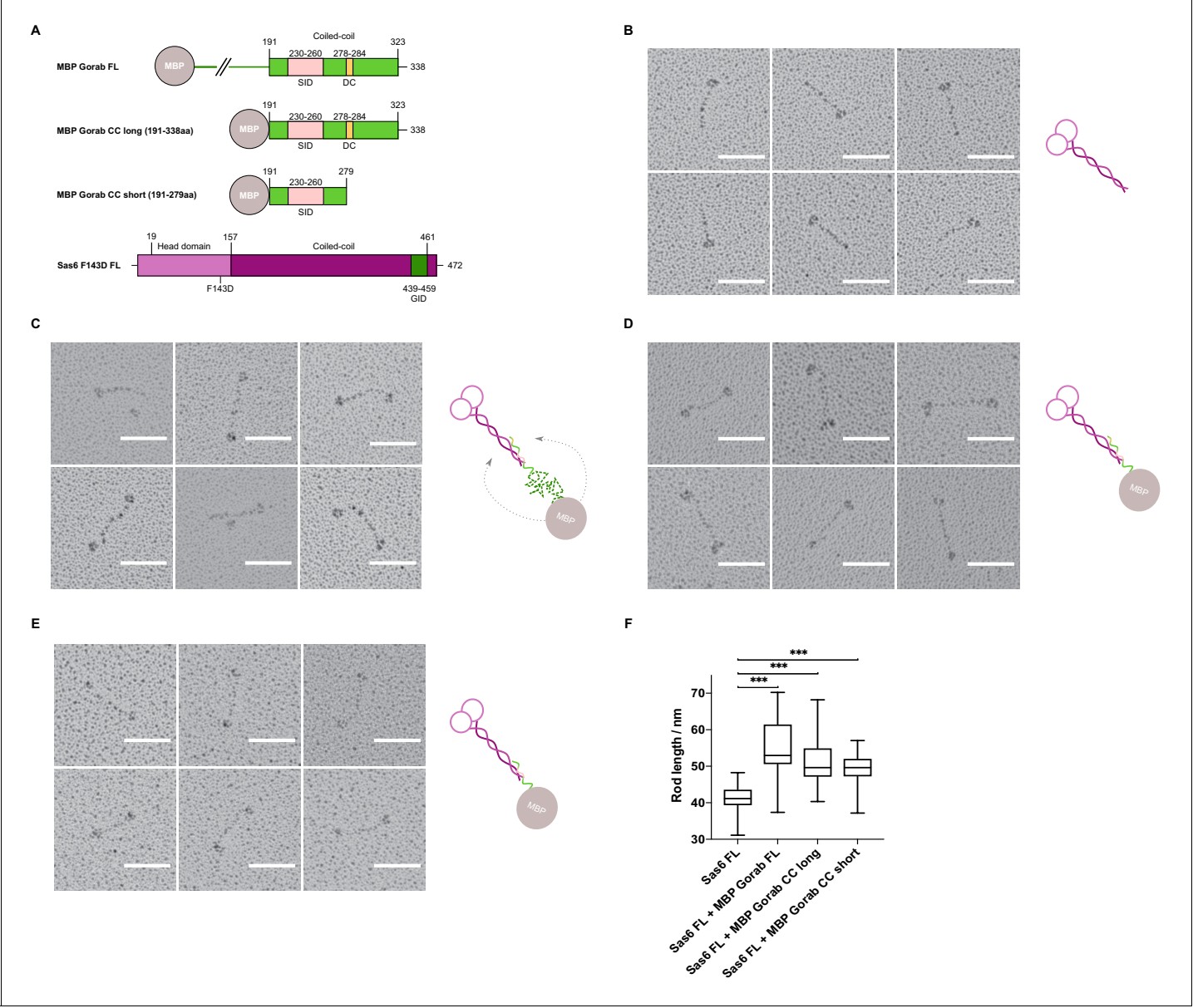

**Figure 5.** Gorab monomer makes an antiparallel interaction with the Sas6 dimer. (A) Schematic depiction of Gorab and Sas6 constructs used for rotary shadowing electron microscopy in B–F. All Gorab constructs have an N-terminal MBP tag. Sas6 has an N-terminal His-tag. The Sas6 F143D mutation disables head-to-head interactions between Sas6 dimers and hence prevents formation of higher oligomeric structures of Sas6. SID: Sas6-interacting domain; DC: dimerization core; GID: Gorab-interacting domain as mapped by hydrogen–deuterium exchange in conjunction with mass spectrometry. (A) Selected electron micrographs of rotary shadowed Sas6. (C) Sas6 in complex with MBP-Gorab full-length (FL). (D) Sas6 in complex with MBP-Gorab 'CC long' (aa 191–338). (E) Sas6 in complex with MBP-Gorab 'CC short' (aa 191–279). Schematic interpretations of structures are shown on the right in each case. Scale bars: 50 nm. (F) Box-and-whisker plots depicting coiled-coil lengths for Sas6 (average length 41.06 nm, n = 106), Sas6 in complex with MBP-Gorab FL (54.85 nm, n = 87), Sas6 in complex with MBP-Gorab 'CC long' (51.51 nm, n = 118), and Sas6 in complex with MBP-Gorab 'CC short' (49.56 nm, n = 114).

The online version of this article includes the following source data and figure supplement(s) for figure 5:

**Source data 1.** Table of rod lengths measured in rotary shadowing micrographs of each of the proteins mentioned.

**Figure supplement 1.** Representative electron micrographs of rotary shadowed molecules.

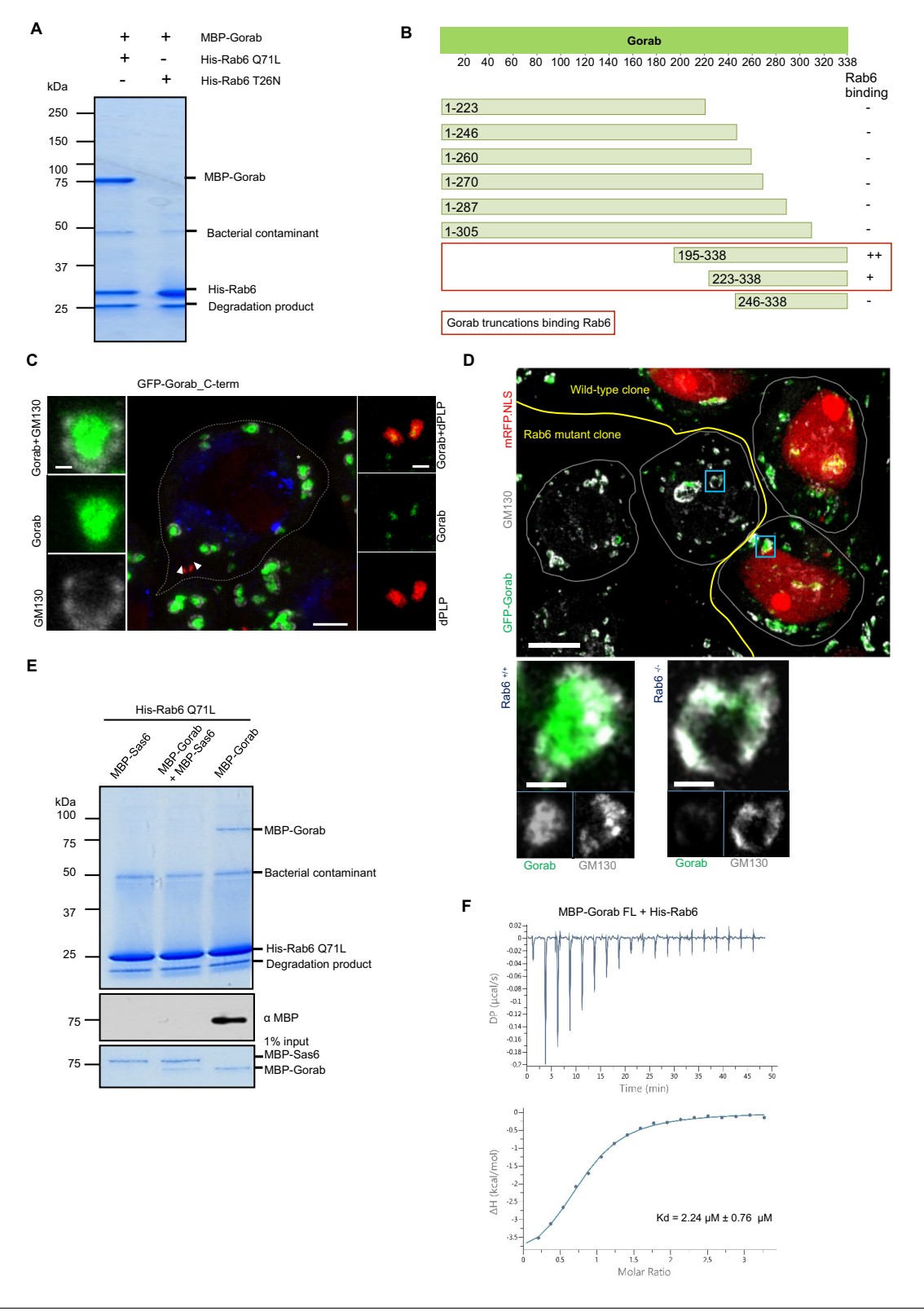

**Figure 6.** Gorab interacts with Rab6 via its C-terminal domain. (**A**) Binding assay for His-Rab6 Q71L (GTP-locked) and His-Rab6 T26N (GDP-locked) with MBP-Gorab. SDS-PAGE shows Rab6 Q71L or Rab6 T26N as bait and Gorab as prey. Gorab binds only the active (GTP-locked) form of Rab6. (**B**) Schematic showing binding of Gorab truncations to Rab6 in vitro. −: no binding; +: binding; ++: strong binding. Red box: region of Gorab essential for Rab6 binding. Individual binding results for each construct are presented in *Figure 6—figure supplement 1B*. (**C**) Localization of C-terminal 'half' (CTH, *Figure 6 continued on next page*

*Figure 6 continued*

aa 195–338) of Gorab in interphase primary spermatocytes of adult testes Primary spermatocytes expressing N-terminally GFP-tagged CTH-Gorab by a poly-ubiquitin promoter were stained to reveal GM130 (Golgi marker, white) and dPLP (centrosome marker, red). Asterisks: Golgi bodies; arrowheads: centrosomes shown in insets. In total, 30 primary spermatocyte were imaged, all showing similar localization. Scale bar: 5 μm; scale bar in insets: 1 μm. (D) Golgi localization of Gorab depends on Rab6. Confocal micrographs of mutant mosaic spermatocytes generated by FLP-FRT recombination in *rab6* heterozygous males expressing GFP-Gorab, stained for Golgi marker GM130. Rab+/+ (red nuclei) and Rab6-/- (nuclei not labeled) cells are indicated. Scale bar: 10 μm; inset scale bar: 1 μm. (E) SDS-PAGE and western blot of in vitro binding assay for His-Rab6 (bait) with MBP-Gorab, MBP-Sas6, or complex MBP-Gorab-MBP-Sas6 (prey). Upper panel: SDS-PAGE showing His-Rab6 Q71L is not able to bind Gorab when Gorab is in the complex with Sas6. Central panel: western blot revealing Gorab; lower panel: SDS-PAGE of the Gorab, Sas6, and Gorab–Sas6 input. (F) ITC profile of His-Rab6 Q71L interacting with MBP-Gorab FL WT. Indicated Kd with standard deviation is the average value of the ITC experiment performed in triplicate.

The online version of this article includes the following figure supplement(s) for figure 6:

**Figure supplement 1.** Regions of Gorab required to interact with Rab6.
**Figure supplement 2.** Deuteration time course identifying regions of Gorab that interact with Rab6.
**Figure supplement 3.** Sas6 interferes with binding of Gorab to Rab6.

As an alternative approach to narrow down the Rab6-interacting region of Gorab, we implemented HDX-MS. Our findings (*Figure 6—figure supplement 2*) are consistent with the binding assays using truncations. The resulting difference spectra between Gorab alone and Gorab bound to Rab6 over intervals between 10 s and 30 min indicated the interaction to be made by the 190–320 aa segment of Gorab, the coiled-coil region of its C-terminal part. However, the precise amino acid region varied as the period of incubation in deuterated water increased. The shorter incubation times of around 10 s identify regions of intermediate stability, around 220–250 aa, whereas at longer times this region is fully exchanged and this difference no longer observed. Instead, at longer times we also localized changes in the more stable 250–300 aa region. HDX therefore indicates destabilization of the whole 220–300 aa region encompassing destabilization both of less and of more stable sections.

We were, however, able to show that the C-terminal Rab6-binding domain of Gorab was sufficient for its Golgi localization by expressing the GFP-tagged C-terminal domain (aa 195–338) alone in flies and demonstrating that it could localize adjacent to GM130 at the Golgi and alongside dPLP at the centriole (*Figure 6C*). As *rab6* mutants are early embryonic lethal, we had to generate homozygous *rab6-/-* clones in a *rab6 +/-* background to be able to examine the requirements for Rab6 for Gorab localization in the fly. Whereas Gorab localized to the Golgi alongside GM130 in *rab6+/-* cells, cells from the *rab6-/-* clone had no Gorab at the Golgi (*Figure 6D*). Together, this strongly suggests that interactions between the dimerizing C-terminal part of Gorab and Rab6 are required for Gorab to localize to the Golgi.

We then considered how Sas6 might interfere with the binding of Gorab to Rab6. To this end, we immobilized Rab6 as bait and then asked if Gorab would bind to it when it is involved in the interaction with Sas6 (*Figure 6E, Figure 6—figure supplement 3*). We showed that Rab6 binds Gorab only as a dimer, but is not able to interact with Gorab when in complex with Sas6 (*Figure 6—figure supplement 3*). This result strongly suggests that the Gorab's interaction with Sas6 is much stronger than its interaction with Rab6. We were able to confirm this using isothermal titration calorimetry (ITC), which revealed that the Gorab–Rab6 complex showed a $K_D$ of 2.24 μM. Thus, the interaction between Gorab and Rab6 is significantly weaker than that between Gorab and Sas6 ($K_D$ = 47 nM) (*Figure 6F*).

A stronger interaction of Gorab with Sas6 than with itself and Rab6 leads to the prediction that increased expression of Sas6 should lead to a reduction in the amount of Gorab associating with the Golgi apparatus. Indeed, we were able to see this in the experiments described above (*Figure 4B*). The elevated expression of both wild-type Sas6 and Sas6-M440A in a *sas6* background had the effect of increasing the GFP-Gorab signal intensity at the centrosomes and reducing it at the Golgi. By contrast, elevated expression Sas6-L447A in the *sas6* null background did not affect the Golgi pool of Gorab, which was not recruited to residual centrosomes, that could have robust staining of Asl (*Figure 4B*). Thus, it seems that the relative levels of Gorab at the centriole and Golgi are set by its high affinity to Sas6 and the concentration of Sas6 at the centriole.

## Discussion

Together, our findings indicate that Gorab exists at the trans-Golgi network as a homodimer. Dimerization requires its coiled-coil motif (residues 200–315) within which is a core sequence (residues 270–287) that represents the most stable part of this dimerization region. Dimerization enables Gorab to interact with Rab6, and this in turn enables its association with the trans-Golgi. In contrast, Gorab interacts with Sas6 as a monomer. Gorab's binding to Sas6 occurs with a higher affinity than its homodimerization, enabling a Gorab monomer to associate with the Sas6 dimer. Thus, the relatively small number of Sas6 molecules at the centriole would more avidly bind the Gorab monomer, allowing greater excess of Gorab to accumulate as dimers at the trans-Golgi (*Figure 7*). Sas6 and Gorab interact through short interfaces within their coiled-coil regions. Disruption of this region of Sas6 through mutation of a single conserved leucine residue, L447, results in a failure of Gorab to bind to Sas6 and localize to the centriole. While we cannot formally exclude the possibility that the L447A mutation affects some other aspect of Sas6 function, the finding that expression of this mutant phenocopies a strong *gorab* hypomorph in its effects upon both co-ordination and centriole duplication suggests that failure to recruit Gorab is responsible for the Sas6-L447A defect. The finding of some residual apparent Gorab-like function in Sas6-L447A-expressing flies may reflect the overexpression of the protein due to the technical requirements of the experiment and the fact that Sas6-L447A still binds Gorab but with a sixteenfold reduced affinity compared to wild-type Sas6. Given that Sas6-L447A greatly diminishes the interaction with Gorab, whereas the mutation, M440A, in the adjoining 'a' position of the 'a–g' coiled-coil heptad repeat does not, leads us to conclude that Gorab binds to a narrow region near the C-terminus of the coiled coil of Sas6.

Gorab shows many of the properties typical of golgins, a family of tentacle-like proteins that protrude from the Golgi membranes to capture a variety of target vesicles. Redundancy between golgins in their ability to bind target vesicles could act as a functional safeguard and might explain why

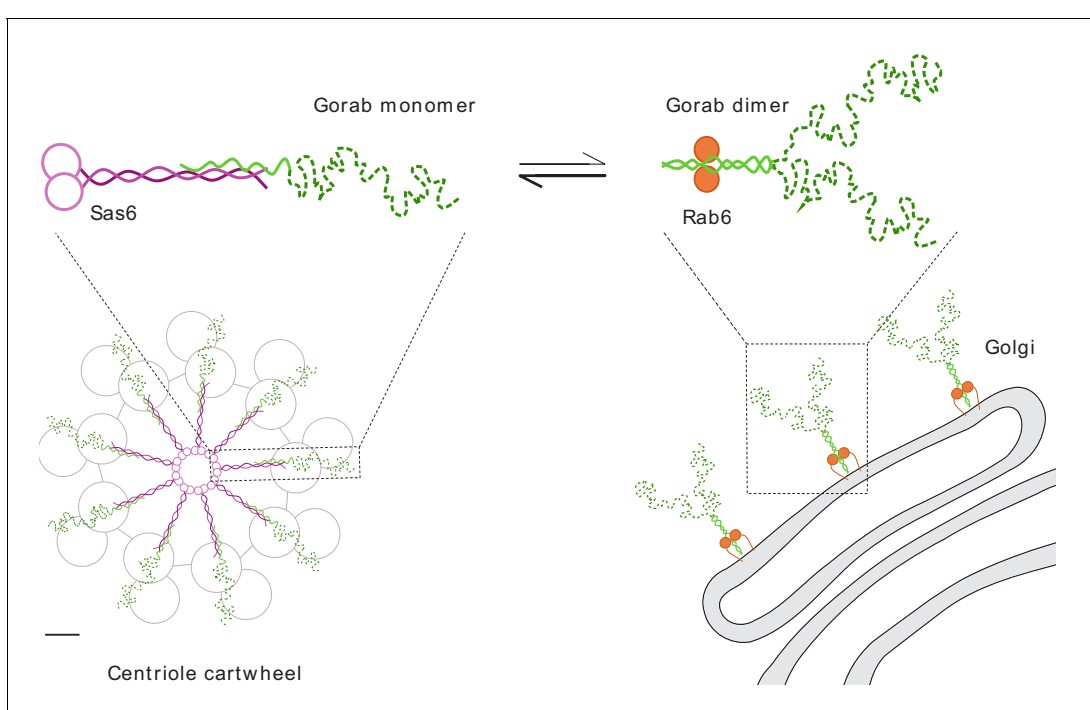

**Figure 7.** Schematic representation of Gorab, its interacting partners Sas6 and Rab6, and the cellular localization of their complexes. Gorab is shown in green: N-terminal dynamic domain in dashed dark green and C-terminal putative coiled-coil domain in solid light green. Sas6 is shown in magenta, depicting two N-terminal head domains and the coiled coil formed in a Sas6 dimer. Rab6 is shown in orange. Although Gorab forms a dimer in solution, it dissociates to a monomer upon binding to Sas6 via its C-terminal coiled-coil domain. The Sas6–Gorab complex present in the centriole cartwheel is shown superimposed over the outer microtubule wall of *Drosophila* centriole consisting of MT doublets, roughly in scale (scale bar 20 nm). Gorab also interacts with Rab6 via its C-terminal coiled-coil domain, but only as a dimer and with a much weaker binding affinity compared to Sas6 binding. Interaction with Rab6 enables Gorab's localization to the trans-Golgi network, similarly to other golgins.

loss-of-function *gorab* mutants display no obvious Golgi phenotype, contrasting to the Golgi defects shown by the C-terminally tagged Gorab molecule (*Kovacs et al., 2018*). Gorab is similar to other golgins, which also associate with the Golgi membranes through their C-terminal parts in interactions that require Rab family member proteins to interact with the C-terminal part of the golgin dimer (*Gillingham and Munro, 2016*). The N-terminal parts of the golgins interact with their vesicle targets. Human GORAB's N-terminal part interacts with Scyl1 to promote the formation of COPI vesicles at the *trans*-Golgi (*Witkos et al., 2019*). However, its precise role in the transport of COPI vesicles is not clear, particularly why loss of human GORAB affects Golgi functions in just bone and skin when COPI function is required in multiple tissues. *Drosophila* Gorab also co-purifies and physically interacts with both Yata, counterpart of Scyl1, and COPI vesicle components (our unpublished data), and its importance for transport of COPI vesicles in *Drosophila* is similarly unclear.

Our study offers a perspective on how Gorab interacts with Sas6 at the centriole and suggests the possibilities for why, as we previously showed (*Kovacs et al., 2018*), this interaction is essential to establish the centriole's ninefold symmetry (*Figure 7*). The heterotrimeric structure formed by a Sas6 dimer and the Gorab monomer will together constitute a single *spoke plus central hub* unit of the centriole's cartwheel. The C-terminal part of Gorab would be expected to lie in a tight antiparallel association with the C-terminal part of Sas6's coiled-coil region. Gorab's N-terminus might thus be expected to extend towards the centriolar microtubules and their associated proteins (*Figure 7*). As the microtubules of *Drosophila's* somatic centrioles exist as doublets of A- and B-tubules, it is tempting to speculate that Gorab interacts with the centriole wall in a region occupied in other cell types by the C-tubule. This could account for the lack of any requirement for Sas6–Gorab interaction in the male germ-line, where centrioles have triplet microtubules and a C-tubule occupies this space. Gorab's partner proteins interacting with its N-terminal region are therefore of great interest at both the Golgi and in the centriole, and it will be key to understand the nature of these interactions in future studies.

## Materials and methods

### Plasmids

All expression vectors were generated using the Gateway system (Invitrogen). The QuickChange Mutagenesis Kit (Agilent) was used to introduce all amino acid-substitution mutations. The constructs were verified by DNA sequencing.

### Protein expression and purification

Recombinant proteins were expressed in *E. coli* strain *BL21(DE3)* (Thermo Fisher) or *Rosetta(DE3)* (Thermo Fisher) following standard procedures. Briefly, bacteria were transformed with recombinant plasmids encoding the desired proteins and cultured at 37°C to $A_{600}$ of approximately 0.5–0.7 in Terrific Broth or Luria Broth supplemented with appropriate antibiotics. Protein expression was induced with 0.5 mM isopropyl-b-D-1-thiogalactopyrano-side at 20°C overnight. Bacterial cells were harvested, resuspended in buffer A (20 mM Tris-HCl pH 7.5, 150 mM NaCl, 5% [v/v] glycerol, 1 mM dithiothreitol [DTT]) supplemented with EDTA-free protease inhibitor cocktail (Roche) and 0.1 mg/ml lysozyme (Sigma-Aldrich) and incubated on ice for 30 min. Cells were lysed by sonication and clarified by centrifugation at 15,000 *g* for 15 min at 4°C. The cleared lysates were incubated with amylose resin (NEB), glutathione sepharose 4B resin (GE Healthcare), or Ni-NTA (Thermo Fisher) resin for MBP-, GST-, or His-tagged proteins, respectively, for 2 hr at 4°C. Beads with bound proteins were washed three times for 10 min with 30 column volumes of buffer A. Bound proteins were eluted with buffer A supplemented with 20 mM maltose, 10 mM glutathione, or 200 mM imidazole.

### SEC and SEC-MALS

For SEC, we used Superose6 10/300 (GE Healthcare) or Superdex 200 10/300 (GE Healthcare) columns pre-equilibrated with buffer A (20 mM Tris-HCl pH 7.5, 150 mM NaCl, 5% [v/v] glycerol, 1 mM DTT). Affinity-purified protein samples were loaded onto the columns and SEC was run at a 0.5 ml/min flow rate at 4°C. Elution of proteins was monitored at 280 nm. Fractions were collected and analyzed by SDS-PAGE and PageBlue protein staining (Thermo Fisher). For HDX-MS studies, the principal fractions having the highest protein concentration were used. SEC-MALS analysis was performed

using a high-performance liquid chromatography (HPLC) instrument (1260 Infinity LC, Agilent Technologies) equipped with a UV detector; samples were monitored at wavelengths of 280, 254, and 215 nm. The HPLC instrument was connected to in-line detectors: a MALS detector (DAWN HELEOS II, Wyatt Technology) and a differential refractometer (Optilab T-rEX, Wyatt Technology). One hundred microliters of protein samples were loaded onto a Superdex 200 Increase 10/300 column (GE Healthcare) or Superose6 Increase 10/300 column (GE Healthcare) equilibrated with buffer A. Samples were run at room temperature at a flow rate of 0.5 ml/min. The results were analyzed using ASTRA v. 6 software (Wyatt Technology) according to the manufacturer's instructions.

## Hydrogen–deuterium exchange mass spectrometry (HDX-MS)

Peptide lists were established by diluting 5 µl of each analyzed protein tenfold into a non-deuterated buffer (20 mM Tris-HCl pH 7.5, 150 mM NaCl, 1 mM DTT). The sample (50 µl) was acidified by mixing with 10 µl of 'stop' buffer (2 M glycine pH 2.5, 1.5 M urea, 250 mM TCEP) and digested offline in the ThermoMixer (Eppendorf) for 30 s at 1°C with 2 µl of protease (*Aspergillus saitoi*, type XIII; Sigma) and then injected into a nanoACQUITY UPLC system (Waters) equipped with an HDX Manager system (Waters) with the column outlet coupled directly with a SYNAPT G2 HDMS mass spectrometer followed by online digestion using an immobilized pepsin column (Porozyme, ABI) with 0.07% formic acid in water as the mobile phase (flow rate 200 µl/min). Digested peptides were trapped on a C18 column (UPLC BEH C18 Van-Guard Pre-column 1.7 µm, 2.1 × 5 mm, Waters) and then directed into a reverse phase column (UPLC BEH C18 column 1.7 µm 2.1 × 50 mm, Waters) with a 10–35% gradient of acetonitrile in 0.1% formic acid at 90 µl/min using nanoACQUITY Binary Solvent Manager. Total time of a single run was 12 min. All capillaries, valves, and columns were maintained at 0.5°C inside an HDX cooling chamber, while the pepsin column was kept at 20°C inside the temperature-controlled digestion compartment. Leucine–enkephalin solution (Sigma) was used as a Lock mass. For protein identification, mass spectra were acquired in MSE mode over the m/z range of 50–1950. The spectrometer parameters were as follows: ESI positive mode, capillary voltage 3 kV, sampling cone voltage 35 V, extraction cone voltage 3 V, source temperature 80°C, desolvation temperature 175°C, and desolvation gas flow 800 l/hr. Peptides were identified using ProteinLynx Global SERVER (PLGS) software (Waters). The list of identified peptides containing peptide m/z, charge, and retention time was further processed with the DYNAMX v. 3.0 program (Waters). For HDX experiments, protein samples were diluted in the reaction buffer containing 99.8% $D_2O$ (Cambridge Isotope Laboratories). Five microliters of protein stock solution was mixed with 45 µl $D_2O$ reaction buffer, and an exchange reaction was carried out for a specific time period (either 10 s, 1 min, 5 min, 30 min, or 150 min) at room temperature (RT) for Gorab alone and Gorab–Rab6 samples and on ice for Gorab, Sas6, and Gorab–Sas6 samples. The exchange was quenched by reducing the pH to 2.5 by adding the reaction mixture into an Eppendorf tube containing ice-cold stop buffer (2 M glycine pH 2.5, 1.5 M urea, 250 mM TCEP). Immediately after quenching, samples were snap-frozen in liquid nitrogen and stored at −80°C until analyzed. Quenched samples were rapidly thawed, digested offline with 2 µl of protease for 30 s at 1°C with shaking, and manually injected into the nanoACQUITY UPLC system. Further digestion, LC, and MS analysis were carried out exactly as described for the non-deuterated sample. For out-exchange control analysis of Gorab samples, measuring the maximum exchange for a given peptide, the 5 µl protein stock was mixed with 45 µl of $D_2O$ reaction buffer, incubated for 24 hr at RT, mixed with stop buffer, and analyzed as described above. The deuteration level in the out-exchange control experiment was calculated and denoted as 100% exchange (Mex100). For protein complexes (Gorab–Sas6 and Gorab–Rab6) to measure the differences in exchange level between the complex samples and proteins alone as out-exchange control, maximal theoretical exchange was used in the calculations. HDX experiments were repeated at least three times. Experiments were repeated using either different overexpression batches (biological replicates) or the same batch (technical replicates).

## Data analysis

A peptide list was created for each protein using the DynamX 3.0 software based on PLGS peptide identifications, with the following acceptance criteria: minimum intensity threshold, 3000; minimum fragmentation products per amino acids for precursor, 0.3 or 0.25; minimum score, 7.5; maximum mass difference between measured and theoretical value for parent ions, 10 ppm. Analysis of the

isotopic envelopes in DynamX 3.0 software was carried out using the following parameters: retention time deviation ±18 s; m/z deviation ±15 ppm; drift time deviation ±2 time bins. Centroids of the mass envelopes were obtained. The values reflecting the experimental mass of each peptide in all possible states, replicates, time points, and charge states were exported from the DynamX 3.0, and further data analysis was carried out using in-house scripts written in R (http://www.R-project.org). The deuterated fraction (D) was calculated with the following formula: D = (Mex – Mex0)/(Mex100 – Mex0), where (Mex0) indicates the average peptides mass with 0% exchange and (Mex100) indicates the average peptide mass measured in out-exchange control, respectively. Error bars for fraction exchanged represent standard deviations calculated from independent replicates. The difference in the fraction exchanged (Δ deuterated fraction) was calculated by subtracting the fraction-exchanged values for peptides in the selected state from the values for the same peptides in the control state. The error bars were calculated as the square root of the sum of the variances from compared states. Student's t-test for independent measurements with unequal variances and unequal sample sizes (also known as Welsh t-test) was carried out to evaluate differences in fraction exchanged between the same peptides in two different states. In the text, the amino acid start- and endpoint of structural or interacting regions identified by HDX-MS is estimated from the positions of peptides identified by MS and presented on the graphs.

## In vitro pull-down assay (binding assay)

In vitro pull-down assays were carried out by incubating the lysate containing bait GST- or His-tagged protein on Glutathione-Sepharose 4B (GE Healthcare) or His-Pur cobalt resin (Thermo Fisher), respectively. After mixing by rotation for 1 hr at 4°C, the beads were washed three times for 10 min with buffer A (20 mM Tris-HCl pH 7.5, 250 mM NaCl, 5% [v/v] glycerol, 1 mM DTT, 0.5% [v/v] Triton). Next, the prey MBP-tagged protein was added and the mixture was incubated for 1 hr at 4°C, followed by 3 × 10 min washes with buffer A. The proteins were eluted by boiling in Laemmli sample buffer and analyzed by SDS-PAGE with PageBlue protein staining (Thermo Fisher). For proteins at lower concentrations, western blot analysis was conducted using anti-MBP antibody.

## Fly stocks

Fly stock $w^{1118}$ was used as wild-type control. All flies in the described experiments were maintained on standard *Drosophila* medium and at 25 °C unless otherwise indicated.

For the rescue and localization studies, both wild-type and mutant Gorab transgenes were cloned into pUWG, whilst the Sas6 WT and mutant transgenes were cloned into pPFW *Drosophila* Gateway (Thermo Fisher Scientific) destination vectors using the Gateway LR Clonase II enzyme mix (Thermo Fisher Scientific). The destination vectors were modified by addition of an attB site as described in *Kovacs et al., 2018*. Transgenic flies were generated by ΦC31 integrase-mediated cassette exchange integrating all Gorab transgenes into attP2 (third chromosomal) genomic locus and all Sas6 transgenes into attP40 (second chromosomal) genomic locus. Mutant lines w; $gorab^1$ and w; $gorab^2$ [13] and w; $sas6^{c02901}$ [27] were used in the rescue experiment with the combination of the generated transgenes. The expression of pUASp-FLAG-Sas6 wild-type or point mutant transgenes was induced with *Act5C-GAL4* ubiquitous or *elav-GAL4* pan-neural drivers, respectively.

To generate *rab6* null mutant mosaics by FLP-FRT site-directed recombination system, $hsFLP^{12}$ yw; *Ubi-mRFP.NLS,FRT40A/CyO* flies were crossed to $rab6^{D23D}$,FRT40A/CyO; *Ubq-GFP-Gorab$^{wt}$/ TM6B* flies. The $hsFLP^{12}$ y w/Y; *Ubi-mRFP.NLS,FRT40A/ $rab6^{D23D}$,FRT40A* progeny were heat shocked (37°C, 1 hr) three times: the first time, 72 hr after hatching, and the subsequent two times, 24 and 48 hr later, respectively. Testes were dissected (see below) from freshly eclosed males and imaged. $rab6^{D23D}$ mutant clones were identified by the absence of nuclear mRFP.NLS signal.

## Detailed genotypes of fly stock used in this study

- $w^{1118}$; $gorab^1$/TM6B, Tb Hu[18] ($gorab^1$/$gorab^1$ homozygotes referred to as *gorab hypomorph in this study*)
- $w^{1118}$; $gorab^2$/TM6B, Tb Hu[18] ($gorab^2$/$gorab^2$ homozygotes referred to as *gorab null in this study*)
- $w^{1118}$; PBac{PB}Sas-6$^{c02901}$ (Sas-6$^{c02901}$/ Sas-6$^{c02901}$ referred to as *sas6 null* in this study)
- w*; Rab6$^{D23D}$/CyO; ry506

- *hsFLP$^{12}$ yw; Ubi-mRFP.NLS,FRT40A/CyO*
- *rab6$^{D23D}$,FRT40A/CyO; Ubq-GFP-Gorab$^{wt}$/TM6B, Tb Hu*
- *w$^{1118}$; Ubq-GFP-Gorab$^{wt}$*
- *w$^{1118}$; Ubq-GFP-Gorab$^{Δ282-286}$*
- *w$^{1118}$; Ubq-GFP-Gorab$^{C-terminal\ half\ (CTH,\ aa\ 195-338)}$*
- *w$^{1118}$; pUASP-FLAG-Sas6$^{wt}$*
- *w$^{1118}$; pUASP-FLAG-Sas6$^{L447A}$*
- *w$^{1118}$; Ubq-GFP-Rcd4* (Panda et al., 2020)
- *P{w+mC.hs = GawB}elav$^{C155}$ (elav-GAL4)*
- *y(1) w[*]; P{w[+mC]=Act5C-GAL4}25FO1/CyO, y[+] (Act5C-GAL4)*

## Cell lines

D.Mel-2 *Drosophila* cell line was initially purchased from Gibco 20 years ago and maintained in the Department of Genetics, University of Cambridge. D.Mel-2 cells were validated by proteomic profiling and microscopy. The fact that D.Mel-2 cells are able to grow in serum-free media helped us to make sure we are working with the right cell line. *Drosophila* D-Mel cells were tested as mycoplasma free upon receipt from the supplier.

## Microscopy

Wing imaginal discs were dissected from third instar wandering larvae and fixed in 4% formaldehyde for 30 min, followed by three 20 min 0.1% PBSTx washes. Samples then were blocked for 1 hr in 10% FBS (in 0.1% PBSTx) followed by 1 hr incubation in chicken anti-dPLP antibody (1:500, raised in house; Qiao et al., 2012) or rabbit anti-Asl antibody (1:800, raised in house; Dzhindzhev et al., 2010). After three 20 min 0.1% PBSTx washes, the discs were incubated with Alexa568 goat-anti-chicken (1:400) or Alexa568 goat anti-rabbit (1:400) and Alexa647 goat anti-chicken (1:300) secondary antibodies for 1 hr. After incubation, samples were washed three times for 20 min in 0.1% PBSTx, then mounted in Vectashield with DAPI.

For primary spermatocyte preparations, testes were dissected from pharenocephalic pupae in PBS. Testes were transferred into 10 µl 8% formaldehyde (in PBS) droplet on a microscope slide, gently squashed with a 22 × 22 mm coverslip, and incubated for 10 min. After incubation, slides were frozen in liquid nitrogen, and coverslips removed before transfer into ice-cold absolute EtOH for 10 min. After three 20 min 0.1% PBSTx washes, samples were blocked in 10% FBS for 1 hr, followed by 1 hr incubation in primary antibodies chicken anti-dPLP (1:500) and rabbit anti-GM130 (1:500, Abcam, #ab30637). After three 20 min 0.1% PBSTx washes, samples were incubated in Alexa568 goat-anti-chicken (1:400) and Alexa647 goat-anti-rabbit (1:400) secondary antibodies for 1 hr. After three 20 min 0.1% PBSTx washes, samples were rinsed once in PB and then mounted in Vectashield+DAPI.

D.Mel cells were transiently transfected with Ubq-Sas6-wt-GFP or Ubq-Sas6-L447A vectors for 48 hr. Cells were fixed with formaldehyde and immunostained as described above in larval wing discs.

All microscopic preparations were imaged using a Leica SP8 confocal laser scanning microscope and processed with ImageJ.

## Analysis of eclosion and coordination

To evaluate the eclosion rate of *gorab* and *sas6* mutants and rescued flies, early pupae from low-density vials were transferred with a wet and soft paintbrush into a fresh vial (40 pupae/vial). After 12 days, the number of dead pupae, eclosed flies, and flies stuck to food was recorded.

To assay coordination through climbing ability, flies were transferred without anesthesia into a clear empty testing vial. Vials were illuminated from above, flies were gently tapped down to the bottom of the vial, and then given 15 s to climb up the vial. Numbers of flies crossing the 5 cm mark were then recorded. The climbing assay was repeated three times for each cohort with similar results. Three independent cohorts of 15 flies were tested per genotype.

Data collected from eclosion and coordination assays were analyzed with two-tailed, unpaired *t*-test, initially in Microsoft Office Excel (version 2007) and subsequently verified by GraphPad Prism (version 5.01). p-Values for each analysis are indicated in corresponding figure legends; a 99% confidence interval was applied in all statistical tests.

## Rotary shadowing electron microscopy

Purified protein samples were first diluted to a concentration of approximately 0.1 mg/ml using a buffer containing 20 mM Tris pH 7.5, 150 mM NaCl, 1 mM DTT, and subsequently diluted in a 1:1 ratio in the spraying buffer containing 200 mM ammonium acetate and 60% (v/v) glycerol, pH 7.6. After dilution, the samples were sprayed onto freshly cleaved mica chips (Agar Scientific, UK) and immediately transferred into a BAL-TEC MED020 high vacuum evaporator (BAL-TEC, Liechtenstein) equipped with electron guns. While rotating, samples were coated with a 0.7-nm-thick layer of platinum (BALTIC, Germany) at an angle of 4–5°, followed by a 6–7 nm layer of carbon (Balzers, Liechtenstein) at 90°. The replicas obtained were floated off from the mica chips, picked up on 400 mesh Cu/Pd grids (Agar Scientific, UK), and observed in an FEI Morgagni 268D TEM (FEI, The Netherlands) operated at 80 kV. Images were acquired using an 11 megapixel Morada CCD camera (Olympus-SIS, Germany).

## Fluorescence correlation spectroscopy

FCS measurements were carried out on a Zeiss Confocor 1 instrument (Carl Zeiss-Evotec, Jena, Germany) using an argon–ion laser with 488 nm wavelength (LASOS Lasertechnik GmbH, Jena, Germany). The MBP-Gorab construct (aa 191–279) was covalently labeled with Atto488 fluorescent dye (ATTO 488 maleimide, ATTO-TEC, AD 488–41), and the fluorescence autocorrelation function was measured at a constant Gorab concentration of approximately 10 nM. Fluorescence intensities were autocorrelated with a hardware correlator (ALV 5000, ALV, Langen, Germany) and data analyzed with the FCS ACCESS software (Carl Zeiss-Evotec). Each sample was measured consecutively up to 10 times with a data acquisition time of 20 s in a buffer containing 20 mM Tris-HCl pH 7.5, 150 mM NaCl, and 1 mM TCEP. Three independent measurements were globally fitted in Prism 8 with a Boltzmann sigmoidal fit and normality of residuals tested with D'Agostino–Pearson omnibus (K2) test.

## Isothermal calorimetry titration

ITC experiments were carried out using MicroCal PEAQ-ITC (Malvern). Ligand (His-Rab6) and cell proteins (MBP-Gorab FL WT and MBP-Gorab FL Δ282–286) were dialyzed in the same buffer (20 mM HEPES pH 7.5, 150 mM NaCl) prior to the measurements. Ligand was used in the concentration range 241–300 μM and cell protein 13–30 μM. Titration settings were as follows: 1 injection of 0.2 μl followed by 18 injections of 2 μl, 180 s spacing time between injections, 750 rpm stirring speed, at 25°C. The resulting data was analyzed using the PEAQ-ITC Analysis software.

## Acknowledgements

We thank all members of the Glover, Dadlez, Dong, and Djinovic labs for their insightful comments and discussions. DMG thanks the Wellcome Trust for an Investigator Award and the National Institute of Neurological Disorders and Stroke of the National Institutes of Health award (R01NS113930). AF and MD were funded in part by the National Science Centre MAESTRO project (UMO-2014/14/A/NZ1/00306), while the instruments were funded in part by the Centre of Preclinical Research and Technology (POIG.02.02.00-14-024/08-00) and the Foundation of Polish Science TEAM-Tech Core Facility (TEAM TECH CORE FACILITY/2016-2/2) grants. Rotary shadowing EM was performed by the EM Facility of the Vienna BioCenter Core Facilities GmbH (VBCF), member of the Vienna BioCenter (VBC), Austria. Work from GD's lab was supported by grant P28231-B28 from the Austrian Science Fund (FWF). ES is an associated member of the Integrative Structural Biology PhD program funded by the Austrian Science Fund (W-1258 Doktoratskollegs).

## Additional information

### Funding

| Funder | Grant reference number | Author |
| --- | --- | --- |
| Wellcome Trust | Investigator Award | Levente Kovacs |
| National Institute of Neurolo- | R01NS113930 | David M Glover |

| | | |
|---|---|---|
| gical Disorders and Stroke | | |
| National Science Centre | MAESTRO (UMO-2014/14/A/NZ1/00306) | Agnieszka Fatalska Michal Dadlez |
| Austrian Science Fund | P28231-B28 | Gang Dong |
| National Institute of Neurological Disorders and Stroke | W-1258 Doktoratskollegs | Emma Stepinac |

The funders had no role in study design, data collection and interpretation, or the decision to submit the work for publication.

### Author contributions
Agnieszka Fatalska, Conceptualization, Formal analysis, Investigation, Visualization, Methodology, Writing - original draft, Writing - review and editing, Project initiation, HDX-MS, Generation of deletions and truncations, in vitro binding assays, SEC-MALS; Emma Stepinac, Formal analysis, Investigation, Visualization, Methodology, Writing - original draft, EM, ITC, FCS, SEC-MALS; Magdalena Richter, Conceptualization, Investigation, Visualization, Methodology, Writing - review and editing, Generation of truncations and point mutations, in vitro binding assays; Levente Kovacs, Visualization, Methodology, Writing - original draft, Writing - review and editing, Formulation and execution of all in vivo studies, Drosophila genetics, immunostaining and confocal microscopy; Zbigniew Pietras, Formal analysis, Assisted AF in SEC-MALS data collection and processing; Martin Puchinger, Formal analysis, Assisted ES in collection and analysis of FCS data; Gang Dong, Michal Dadlez, Conceptualization, Supervision, Funding acquisition, Project administration, Writing - review and editing; David M Glover, Conceptualization, Supervision, Funding acquisition, Writing - original draft, Project administration, Writing - review and editing

### Author ORCIDs
Agnieszka Fatalska https://orcid.org/0000-0002-1720-4742
Gang Dong https://orcid.org/0000-0001-9745-8103

### Decision letter and Author response
Decision letter https://doi.org/10.7554/eLife.57241.sa1
Author response https://doi.org/10.7554/eLife.57241.sa2

## Additional files

### Supplementary files
- Transparent reporting form

### Data availability
All data generated or analysed during this study are included in the manuscript and supporting files.

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
