## [Decision Letter]

**Acceptance summary:**

Previous work has found that a conserved Golgi protein Gorab is required for centriole duplication in flies, and the current study provides important insights into the underlying molecular mechanism. The study will be interesting to cell biologists and biochemists working the cytoskeleton and trafficking.

**Decision letter after peer review:**

Thank you for submitting your article "A heterotrimer of the Gorab trans-Golgi protein and Sas6 is required for centriole duplication" for consideration by *eLife*. Your article has been reviewed by three peer reviewers, and the evaluation has been overseen by a Reviewing Editor and Anna Akhmanova as the Senior Editor. The following individual involved in review of your submission has agreed to reveal their identity: Meng-Fu Bryan Tsou (Reviewer #3).

The reviewers have discussed the reviews with one another and the Reviewing Editor has drafted this decision to help you prepare a revised submission.

As the editors have judged that your manuscript is of interest, but as described below that additional experiments are required before it is published, we would like to draw your attention to changes in our revision policy that we have made in response to COVID-19 (https://elifesciences.org/articles/57162). First, because many researchers have temporarily lost access to the labs, we will give authors as much time as they need to submit revised manuscripts. We are also offering, if you choose, to post the manuscript to bioRxiv (if it is not already there) along with this decision letter and a formal designation that the manuscript is "in revision at *eLife*". Please let us know if you would like to pursue this option.

Summary:

GORAB is an intriguing factor that the Glover lab has shown to display both Golgi and localisation and function in centriole duplication. Importantly, they have shown that these reflect separate functions of GORAB. In this work, the authors examine the structure of GORAB in more detail, mapping the regions important for Golgi targeting, SAS6 binding and centriole duplication in detail. Using this information, they produce a separation of function mutant in SAS6 that cannot bind to GORAB. This L447A mutant has reduced affinity for GORAB in vitro and fails to fully rescue the sas6 mutant, resulting in a phenotype described as similar to that of the gorab mutant. Overall, the authors provide a detailed analysis of GORAB dynamics with some clever use of methodology. This data is also of high quality. However, as detailed below, it appears that further experiments would be needed to bring the findings to the level expected from a paper in *eLife*.

Essential revisions:

All three reviewers agreed that the key question concerns the interpretation of Sas6 L447A mutant. The authors demonstrate that this mutant protein is severely compromised, both in its interaction with Gorab and in its function in *Drosophila*. These observations lead the authors to suggest that the in vivo phenotype of Sas6 L447A is due to impaired interaction with Gorab. In light of the available data, this suggestion seems premature. The authors should investigate the impact of the L447A mutation on Sas6 itself. According to Deepcoil, L447 is likely to be within the coiled coil, such that mutating a Leucine occupying an a or d position within the heptad repeat is expected to affect protein stability. Does Sas6 L447A fold correctly? Can it undergo homodimerization? These questions must be addressed. The authors could potentially investigate how the protein looks by rotary metal shadowing on its own, as well as in the presence of Gorab, either in the wild-type or the mutant form lacking the 244-259 stretch normally mediating interaction with Sas6. The authors should also report the localization of Sas6 L447A, something that is currently not shown in Figure 3D. In the absence of such results, one likely explanation could be that Sas6 L447A protein is compromised.

Even if the Sas6 L447A folding is not compromised, the impact of this mutation may not be caused by the lack of GORAB binding, since another factor could interact with that region. In principle, engineering a compensatory mutation in Gorab that restores interaction with Sas6 L447A would be desirable. If this is not possible, this point needs to be explicitly discussed.

Is SAS6 binding required for GORAB function or vice versa? We assume the latter is the case due to the previous data where a GORAB mutant unable to target to Golgi that could bind SAS6 did rescue centriole phenotypes. In this work, a SAS6 null rescued with SAS6 L447A has a phenotype the authors describe as resembling the Gorab null. A more detailed analysis of centriole duplication in the SAS6 L447A rescue would help to address this point as well as the points above.

1) The authors performed an impressive systematic structure/function analysis to uncover the domains needed for Gorab dimerization (Figure 1D), Sas6 binding (Figure 2D, Figure 2—figure supplement 2A) and Rab6 interaction (Figure 6B, Figure 6—figure supplement 1B). However, the qualitative assessment reported in Figure 2D or Figure 6B does not always match what the gels show in Figure 2—figure supplement 2A or Figure 6—figure supplement 1B. For instance, it seems clear from Figure 2—figure supplement 2A that the delta219-244 or the delta267-281 Sas6 constructs bind Gorab very weakly, yet they are reported as regular binders in Figure 1D. This seems to suggest that a much larger swath of Gorab mediates interaction with Sas6. An analogous comments holds for Rab6: the gel in Figure 6—figure supplement 1B shows clearly that delta223-339 hardly binds Rab6, whereas Figure 6B reports this fragment as a regular binder. These apparent inconsistencies must be clarified with a quantitative assessment.

2) The low magnification views of Figure 1E are difficult to fully understand, as the quality of this immunostaining is not on par with the beautiful distribution of Gorab previously reported by these authors. Moreover, the DNA signal does not appear to be clearly present between what supposedly must be spindle poles. And what does the dashed line delineate? This information is not provided in the figure legend. Moreover, this experiment should be quantified: how many cells were scored and what fraction of them exhibited the illustrated distribution? A related comment holds for Figure 3D, where the contours of individual cells is difficult to make out in addition.

---

## [Author Response]

Essential revisions:All three reviewers agreed that the key question concerns the interpretation of Sas6 L447A mutant. The authors demonstrate that this mutant protein is severely compromised, both in its interaction with Gorab and in its function in *Drosophila*. These observations lead the authors to suggest that the in vivo phenotype of Sas6 L447A is due to impaired interaction with Gorab. In light of the available data, this suggestion seems premature. The authors should investigate the impact of the L447A mutation on Sas6 itself. According to Deepcoil, L447 is likely to be within the coiled coil, such that mutating a Leucine occupying an a or d position within the heptad repeat is expected to affect protein stability. Does Sas6 L447A fold correctly? Can it undergo homodimerization? These questions must be addressed.

To address this question, we have carefully analysed the mutant Sas6-L447A. Our SEC data showed that the L447A mutation does not affect the ability of Sas6 to dimerize. As seen in Figure 3—figure supplement 3B, Sas6-wild type, Sas6-M440A, Sas6-L447A, Sas6-S452A and Sas6-L456A all displayed the same elution profile in size exclusion chromatography. This indicates that all the Sas6 point mutants behave like Sas6-wild type in that they are all able to form homodimers.

We have also modeled the organization of the coiled-coil structure of Sas6 (Figure 3—figure supplement 4). Consistent with the in vitro dimerization of the mutants, our 3D modeling of Sas6-CCD shows a very small change in BUDE energy for both mutants M440A and L447A (-717/-714 vs -724 for WT). Notably, however, all these can only confirm that most of the coiled coil remains intact and intertwined, but cannot exclude a possibility that the local structure around L447 is partially perturbed in the mutant L447A. Both M440 and L447 are predicted to locate at position “a” of the “a-g” heptad repeats, but residue L447 is even closer to the C-terminus of the predicted CCD (i.e. aa464). Additionally, the downstream two heptads after L447 may not hold the dimer tightly together as only one of those residues at positions "a" and "d" in that region are hydrophobic (i.e. I457).

Therefore, we rephrased our original conclusion by stating that “Given that Sas6-L447A greatly diminishes the interaction with Gorab, whereas the mutation, M440A, in the adjoining “a” position of the “a-g” coiled-coil heptad repeat does not, leads us to conclude that Gorab binds to a narrow region near the C terminus of the coiled-coil of Sas6.”

The fact that Sas6-M440A, which like L447 is predicted to locate at position “a” of the “a-g” heptad repeats, does not perturb Gorab binding and rescues the Sas6 null mutation to a similar extent as Sas6 wild type, together with the phenotypic similarity of the Sas6 L447A mutant with gorab mutants suggest that the primary cause of the Sas6 L447A phenotype is inappropriate binding of Gorab rather than disturbance in overall structure of Sas6.

The authors could potentially investigate how the protein looks by rotary metal shadowing on its own, as well as in the presence of Gorab, either in the wild-type or the mutant form lacking the 244-259 stretch normally mediating interaction with Sas6. The authors should also report the localization of Sas6 L447A, something that is currently not shown in Figure 3D. In the absence of such results, one likely explanation could be that Sas6 L447A protein is compromised.

As previously indicated by size exclusion chromatography, Sas6-L447A mutant behaves the same as Sas6-wild type and forms a homodimer. Additional 3D modeling also suggests that the coiled coil of Sas6-L447A is intact and intertwined for the most of its length, although local structural disturbance around L447 position cannot be excluded. Even though rotary shadowing beautifully enables us to visualize both the head domains and the elongated coil of Sas6 molecule, the method has a limited spatial resolution and it is not high enough to visualize local disturbances along the rod of the Sas6 coiled coil, if there are any present at all. Since the coiled coil is predicted to be stable in its length otherwise, we would not see a difference in the rod length in the EM micrographs. From our FCS measurement, It seems unlikely that the Sas6-L447A : Gorab interaction is sufficiently strong to withstand the procedures used to prepare the complex for rotary shadowing and the rotary shadowing procedure itself.

We have carried out transient transfection of cultured *Drosophila* cells with C-terminally tagged Sas6 L447A and now show in Figure 4—figure supplement 1 that all cells expressing the transgene have GFP localized to the centrioles. Thus Sas6-L447A is not only able to dimerise, but also to localize correctly to the centriole.

Even if the Sas6 L447A folding is not compromised, the impact of this mutation may not be caused by the lack of GORAB binding, since another factor could interact with that region. In principle, engineering a compensatory mutation in Gorab that restores interaction with Sas6 L447A would be desirable. If this is not possible, this point needs to be explicitly discussed.

We agree that the fact that Gorab is not properly recruited centrioles doesn’t exclude the possibility that other Sas6 interactors are also not affected and include a sentence to this effect in the revised discussion. However, we now show that Sas6 L447A does not just phenocopy the extent of the uncoordination phenotype of gorab flies (Figure 3D-F), but have now added new data to show that it also mimics the extent of centriole duplication and structural defects, which are reminiscent of a strong gorab hypomorph (Figure 4 and Figure 4—figure supplement 1).

Together, the finding that Sas6-L447A phenocopies the centriolar phenotype of gorab mutants, the presence of Sas6-L447A on centrioles and the absence/very low amount of Gorab on such centrioles strongly suggests that the absence of Gorab is the primary cause of the mutant phenotype.

Is SAS6 binding required for GORAB function or vice versa? We assume the latter is the case due to the previous data where a GORAB mutant unable to target to Golgi that could bind SAS6 did rescue centriole phenotypes.

As indicated, we previously described a series of separation of function alleles – namely the gorab-V266P and various gorab deletion mutations – that completely rescue the centriolar phenotype but are unable to localize to the Golgi. However, these experiments do not tell anything about dependencies because these mutants are able to bind Sas6 normally. We can now say that Sas6 is epistatic over Gorab because: first, the phenotype of the sas6-null mutant is stronger than the gorab null; and second, because Sas-L447A can localize to centrioles, as we now demonstrate (Figure 4—figure supplement 1D), and rescue the sas6-null phenotype to an extent comparable to the phenotype of a strong gorab hypomorphic mutant. Taken together with our finding that there is either no or very little Gorab at centrosomes in the Sas6-L447A mutant (Figure 4C), this demonstrates that Sas6 is needed for Gorab recruitment to the centrosomes. Thus, Sas6 binding is essential for Gorab’s centriolar function.

In this work, a SAS6 null rescued with SAS6 L447A has a phenotype the authors describe as resembling the Gorab null. A more detailed analysis of centriole duplication in the SAS6 L447A rescue would help to address this point as well as the points above.

This is an excellent point, which we have addressed thoroughly in the revised manuscript. We now describe a detailed analysis of both the coordination and centriolar phenotypes of the Sas6 L447A mutant and two gorab mutants. The coordination phenotypes are presented in panels D and E of Figure 3 and the consequences for centrosome organization and number in interphase and mitosis in a new figure (Figure 4). Our findings show that the coordination and centriolar phenotypes of Sas6 L447A resemble those of a strong gorab hypomorph (the heteroallelic combination of a gorab hypomorph and gorab null) rather than the gorab null itself. We discuss the possibility that the phenotype is not quite as strong as the gorab null because Sas6-L447A can still bind a very low amount of Gorab (there is a 16-fold difference in dissociation constant between L447A and WT) and because, due to technical reasons, Sas6-L447A was expressed at higher levels than wild-type Sas6 would be expressed in our experiments.

1) The authors performed an impressive systematic structure/function analysis to uncover the domains needed for Gorab dimerization (Figure 1D), Sas6 binding (Figure 2D, Figure 2—figure supplement 2A) and Rab6 interaction (Figure 6B, Figure 6—figure supplement 1B). However, the qualitative assessment reported in Figure 2D or Figure 6Bdoes not always match what the gels show in Figure 2—figure supplement 2A or Figure 6—figure supplement 1B. For instance, it seems clear from Figure 2—figure supplement 2A that the delta219-244 or the delta267-281 Sas6 constructs bind Gorab very weakly, yet they are reported as regular binders in Figure 1D. This seems to suggest that a much larger swath of Gorab mediates interaction with Sas6.

Thank you for these comments. We did not wish to make any claims about the strength of binding from band intensities that show some variation – we have now attached further examples from additional experiments in the supplements to Figure 2 to illustrate this better. Together these experiments indicate that del 244-259 is the shortest deletion that does not bind in the gel assay. HDX indicates that 230-260 is the region of the strongest interaction. Further support comes from the gel assay of binding of truncations: Trc 1-246 does not bind; Trc 1-260 binds weakly; Trc 1-270 binds like wild-type (please see additions to the supplementary figure). We have attempted to give an indication of the relative binding strength of the various deletions and truncations from repeated experiments, in so much that it is possible from such gel assays, classifying the binding as -, no binding; +, binding ; and ++, strong binding. In summary, 246-260 is the shortest region crucial for binding. However, the sequences in regions 219-244 (del), as well as 260-270 (trc), could also support the interaction while not being essential.

Thus, we conclude that region 246-260 is needed for binding but cannot exclude the possibility that some interactions extend out of this region. We have now clarified this in the main text.

An analogous comments holds for Rab6: the gel in Figure 6—figure supplement 1B shows clearly that delta223-339 hardly binds Rab6, whereas Figure 6B reports this fragment as a regular binder. These apparent inconsistencies must be clarified with a quantitative assessment.

We have clarified the text trying to exclude inconsistencies and to be more cautious in our interpretation. Whereas gel binding assays such as these cannot give accurate quantitative comparisons, they do identify the broad region required for Rab6 binding and the findings we report accord with studies on human Gorab (Egerer et al., 2015)). It is the case that both the Gorab fragments 195-338 and 223- 338 bind Rab6 – the latter is weaker but it is the shortest fragment able to bind Rab6. We label fragment 195-338 on Figure 6B to indicate strong binding, ++, and fragment 223-338 to show binding, +. Together, this accords with the HDX data that shows the entire region C-terminal to residue 200 to change conformation upon Rab6 binding.

2) The low magnification views of Figure 1E are difficult to fully understand, as the quality of this immunostaining is not on par with the beautiful distribution of Gorab previously reported by these authors. Moreover, the DNA signal does not appear to be clearly present between what supposedly must be spindle poles. And what does the dashed line delineate? This information is not provided in the figure legend.

We thank the reviewer for this comment, which brought to our attention the inadequate explanation of this figure in our original manuscript. In our previous paper we described the localization of Gorab in cultured *Drosophila* cells and wing imaginal discs. Here we show Gorab localization in squashed preparations of primary spermatocytes because 1) such tissue can be obtained the first transgenic male from the transformation experiment after it has been crossed to virgin females to establish the stock. Once larvae have been obtained to secure the stock, we immediately dissect and analyze the primary spermatocytes from the founder male; 2) instead of the scattered Golgi stacks of cultured cells and wing disc cells, the primary spermatocytes have larger Golgi bodies with distinct outer (cis-Golgi) and inner (trans-Golgi) layers in cross section (Belloni et al., 2012, Sechi et al., 2013). This gives a clear demonstration of whether the outer ring of GM130 (the cis-Golgi marker) is rounding a compartment that either has or does not have Gorab.

Primary spermatocytes are an extended G2 phase. The spindles have yet to form and the nuclei have a characteristic tri-lobal shape representing the sets of the major chromosome homologues preparing for meiosis. The dashed lines highlight the border of single primary spermatocyte cells in the field of this squashed tissue preparation. We have updated the figure legends for clarification. In addition, we have also imaged GFP-Gorab (full length) and GFP-Gorab Δ282-286 in the mitotic wing disc cells (Figure 1—figure supplement 3).

Moreover, this experiment should be quantified: how many cells were scored and what fraction of them exhibited the illustrated distribution?

We imaged 32 primary spermatocytes and visually observed many more. All of them showed the same Golgi and centriolar distribution of Gorab WT or Gorab Δ282-286. We note that both wild-type and mutant transgenes were integrated at the same genomic landing site (see Materials and methods) so that the transgenes would be expressed at the same level in every animal and every tissue from a poly-ubiquitin promoter. Thus would guard against variability between spermatocytes.

A related comment holds for Figure 3D, where the contours of individual cells is difficult to make out in addition.

We agree that it is difficult to make the contours of the interphase cells in wing disc since these are columnar epithelia and are tightly arranged next to each other. We now present new data in alternative figures (Figure 4 and Figure 4—figure supplement 1) that quantitate the consequences of Sas6-L447A, Sas6-M440A, and different mutant alleles of gorab upon centrioles in diploid tissues.